# The Microtubule-Targeting Agent Pretubulysin Impairs the Inflammatory Response in Endothelial Cells by a JNK-Dependent Deregulation of the Histone Acetyltransferase Brd4

**DOI:** 10.3390/cells12162112

**Published:** 2023-08-21

**Authors:** Tobias F. Primke, Rebecca Ingelfinger, Mohammed A. F. Elewa, Igor Macinkovic, Andreas Weigert, Matthias P. Fabritius, Christoph A. Reichel, Angelika Ullrich, Uli Kazmaier, Luisa D. Burgers, Robert Fürst

**Affiliations:** 1Institute of Pharmaceutical Biology, Goethe University Frankfurt, 60438 Frankfurt, Germany; primke@em.uni-frankfurt.de (T.F.P.); schwenk@em.uni-frankfurt.de (R.I.); burgers@em.uni-frankfurt.de (L.D.B.); 2LOEWE Center for Translational Biodiversity Genomics (LOEWE-TBG), Goethe University Frankfurt, 60596 Frankfurt, Germany; 3Institute of Biochemistry I, Faculty of Medicine, Goethe University Frankfurt, 60596 Frankfurt, Germany; elewa@biochem.uni-frankfurt.de (M.A.F.E.); macinkovic@biochem.uni-frankfurt.de (I.M.); weigert@biochem.uni-frankfurt.de (A.W.); 4Biochemistry Department, Faculty of Pharmacy, Kafrelsheikh University, Kafr El-Sheikh 33516, Egypt; 5Department of Otorhinolaryngology, Walter Brendel Centre of Experimental Medicine, University Hospital, 81377 Munich, Germany; matthias.fabritius@med.uni-muenchen.de (M.P.F.); christoph.reichel@med.uni-muenchen.de (C.A.R.); 6Department of Radiology, University Hospital, University of Munich, 81377 Munich, Germany; 7Institute of Organic Chemistry, Saarland University, 66123 Saarbrücken, Germany; a.ullrich@mx.uni-saarland.de (A.U.); u.kazmaier@mx.uni-saarland.de (U.K.)

**Keywords:** inflammation, endothelium, leukocyte adhesion cascade, cell adhesion molecules, microtubule-targeting agents, c-jun-N-terminal kinase, NFκB, bromodomain-containing protein 4

## Abstract

The anti-inflammatory effects of depolymerizing microtubule-targeting agents on leukocytes are known for a long time, but the potential involvement of the vascular endothelium and the underlying mechanistic basis is still largely unclear. Using the recently synthesized depolymerizing microtubule-targeting agent pretubulysin, we investigated the anti-inflammatory potential of pretubulysin and other microtubule-targeting agents with respect to the TNF-induced leukocyte adhesion cascade in endothelial cells, to improve our understanding of the underlying biomolecular background. We found that treatment with pretubulysin reduces inflammation in vivo and in vitro via inhibition of the TNF-induced adhesion of leukocytes to the vascular endothelium by down-regulation of the pro-inflammatory cell adhesion molecules ICAM-1 and VCAM-1 in a JNK-dependent manner. The underlying mechanism includes JNK-induced deregulation and degradation of the histone acetyltransferase Bromodomain-containing protein 4. This study shows that depolymerizing microtubule-targeting agents, in addition to their established effects on leukocytes, also significantly decrease the inflammatory activation of vascular endothelial cells. These effects are not based on altered pro-inflammatory signaling cascades, but require deregulation of the capability of cells to enter constructive transcription for some genes, setting a baseline for further research on the prominent anti-inflammatory effects of depolymerizing microtubule-targeting agents.

## 1. Introduction

Inflammatory processes defend a host against detrimental factors like pathogens or cellular damage. If not resolved, inflammation gives rise to a multitude of chronic diseases, like rheumatoid arthritis, inflammatory bowel disease, familial Mediterranean fever, multiple sclerosis or even cancer. This eventually leads to a decrease in the patient’s quality of life and a reduced life span [1,2]. The onset of inflammation is characterized by the recruitment of leukocytes from the blood stream and their infiltration of the inflamed tissues. The vascular endothelial cell (EC) layer that lines the inside of small venous blood vessels primarily acts as an interface for the interaction with immune cells. This interaction enables the extravasation of leukocytes from the blood to the inflamed tissue. A prerequisite for this interaction is the pro-inflammatory activation of the endothelial cells by cytokines, like the tumor necrosis factor (TNF), initially secreted by tissue sentinel cells like macrophages or dendritic cells [3,4,5]. TNF exerts its pro-inflammatory function in endothelial cells by inducing the phosphorylation-dependent activation of the IκB kinase (IKK) [6,7]. Activated IKK subsequently phosphorylates members of the inhibitor of nuclear factor kappa B (IκB) family of proteins, which sequester the pro-inflammatory transcription factor nuclear factor kappa B (NFκB) in the cytosol by masking its nuclear localization signal (NLS). Phosphorylation of the IκB proteins by IKK leads to their release from NFκB and subsequent ubiquitination and proteasomal-based degradation. The heterodimeric NFκB transcription factor complex, consisting of the p65 and p50 subunits, then translocates into the nucleus, where it binds the promoters of pro-inflammatory target genes. In addition to NFκB, TNF stimulation also leads to the activation of the c-Jun N-terminal kinase (JNK) family of mitogen-activated protein kinases (MAPKs). JNKs are best known for their downstream activation of the pro-inflammatory transcription factor c-Jun, as a component of the activator protein-1 (AP-1) early response family of transcription factors [8,9]. NFκB and AP-1 act, together with a multitude of other transcription factors, in concert to induce the expression of pro-inflammatory target genes. Considering the extravasation of leukocytes, this comprises the endothelial cell-based upregulation of components of the leukocyte adhesion cascade, like the cell adhesion molecules (CAMs) E-selectin, the intercellular adhesion molecule 1 (ICAM-1), the vascular CAM 1 (VCAM-1) and the platelet endothelial CAM 1 (PECAM-1) [10]. These CAMs enable the different and tightly regulated steps of the leukocyte adhesion cascade: the rolling of the leukocytes over the endothelial cell surface (E-selectin), the firm adhesion of leukocytes (ICAM-1 and VCAM-1) and the subsequent transmigration of the leukocytes through the endothelium (PECAM-1). Due to its critical function during the inflammatory response, the endothelium represents a valuable target for anti-inflammatory compounds. Just recently, the monoclonal antibody drugs natalizumab, vedolizumab and crizanlizumab have been shown to be effective in the treatment of multiple sclerosis, familial Mediterranean fever and vaso-occlusive crisis in sickle-cell disease [11,12,13], respectively. One of the oldest-known groups of anti-inflammatory compounds is the microtubule-targeting agents (MTAs, [14]), with colchicine being a lead example, known for its use in acute gouty arthritis and familial Mediterranean fever [15,16]. Colchicine is mainly known to exert its anti-inflammatory effects by affecting formation of the inflammasome, leukocyte motility via targeting of the microtubule cytoskeleton, as well as the concomitant down-regulation of genes involved in leukocyte migration [17,18,19]. In addition, colchicine has been shown to reduce the infarct-incidence in cardio-vascular diseases and recent research revealed that treatment with colchicine also reduces vascular inflammation and thereby decreases the formation of aortic plaques during atherosclerosis [20,21]. However, while the research on colchicine still leads to new findings concerning the potential usage of this old drug in the treatment of pro-inflammatory diseases, little is known about the mechanistic background of action that MTAs induce in cells of the vascular endothelium.

On this basis, we aimed at assessing the effects of the newly synthesized and very potent microtubule-destabilizing agent pretubulysin [22,23,24,25,26], a chemically accessible precursor of the myxobacterial tubulysins and other MTAs already in clinical use, on pro-inflammatory factors involved in the leukocyte adhesion cascade in human umbilical vein endothelial cells (HUVECs) and to elucidate the underlying mechanisms of MTAs on inflammatory processes of the vascular endothelium.

## 2. Materials and Methods

### 2.1. Compounds

Pretubulysin was synthesized as previously described [27]. Colchicine (CAS 64-86-8), vincristine (CAS 2068-78-2) and paclitaxel (CAS 33069-62-4) were obtained from Biomol (Hamburg, Germany). The compounds were dissolved in DMSO to a concentration of 10 mM and stored at −20 °C. For experiments, compounds were further diluted in cell culture medium prior to application at the concentrations indicated in the figure legends, while not exceeding a final DMSO concentration of 0.01% (*v*/*v*) during treatment in in vitro assays. The inhibitors JNK-IN-8 (CAS 1410880-22-6; stock: 100 mM), staurosporine (CAS 62996-74-1; stock: 1 mM), cycloheximide (CAS 66-81-9; stock: 100 mg/mL) and actinomycin D (CAS 50-76-0; stock 20 mg/mL) were purchased from Sigma-Aldrich (Munich, Germany) and were dissolved in DMSO, before being diluted further in cell culture medium and used at the concentrations indicated in the figure legends.

### 2.2. Cell Culture

Primary human umbilical vein endothelial cells (HUVECs) were either prepared from umbilical veins of healthy donors (waiver W1/21Fü has been granted for the use of anonymized human material on 15 September 2021, issued by the head of the Research Ethics Committee/Institutional Review Board) according to the method of Jaffe et al. [28] or obtained from PeloBiotech (Martinsried, Germany). Cells were seeded in cell culture flasks or plates pre-coated with 10 µg/mL collagen G in PBS and cultivated in endothelial cell growth medium (ECGM; PeloBiotech, Germany) supplemented with 10% fetal calf serum (FCS; PanBiotech, Aidenbach, Germany), 100 U/mL penicillin, 100 µg/mL streptomycin and 2.5 µg/mL amphotericin B (PanBiotech, Germany). The human monocyte-like cell line THP-1 was obtained from the Leibnitz Institute for German Collection of Microorganisms and Cell Cultures (DSMZ, Braunschweig, Germany) and was cultivated in RPMI-1640 (PanBiotech, Germany) containing 10% FCS, 100 U/mL penicillin and 100 µg/mL streptomycin. All cells were cultured at 37 °C in an atmosphere of 5% CO_2_ at constant humidity.

### 2.3. Animals

In vivo experiments were performed using 8- to 10-week-old male or female C57BL/6N/J mice obtained from Charles River Laboratories (Wilmington, MA, USA). Mice were housed based on a 12 h light/dark cycle with free access to food and water.

### 2.4. Psoriasiform Dermatitis Model

The imiquimod (IMQ)-induced psoriasis mouse model was performed and skin severity was assessed as previously described (van der Fits et al. [29]). Mouse back skin was shaved one day before starting the experiment. 62.5 mg commercially available cream containing 5% IMQ (Aldara; 3M Pharmaceuticals, Neuss, Germany) was daily applied on the back skin of 8–10 weeks old male or female mice for 6 consecutive days. Skin severity was evaluated according to the PASI scoring system based on the extent of skin thickness, redness and scaling for up to 10 days. Additionally, the mice received s.c. injections of 1 mg/kg PT or vehicle (PBS) on day 1, 3, and 5. All mouse experiments were approved by and followed the guidelines of the Hessian animal care and use committee (FU/2064).

### 2.5. Intravital Microscopy of the Mouse Cremaster Muscle

The interaction of endothelial cells with leukocytes (Gr-1^+^; neutrophils and classical monocytes) was analyzed in vivo by intravital microscopy of the mouse cremaster muscle. 30 min after intraperitoneal application of pretubulysin (1 mg/kg) or drug vehicle, TNF (25 µg/kg) was injected intrascrotally to induce leukocyte recruitment to the cremaster muscle. After 4 h, mice were anesthetized (ketamine/xylazine) and the right cremaster muscle was prepared as previously described by Baez et al. [30,31]. Intravital microscopy of the cremaster muscle was performed using an Olympus BX 50 upright microscope (Olympus, Tokyo, Japan). Analysis was conducted with the Cap-Image analysis software (Dr. Zeintl, Heidelberg, Germany). All experiments were approved by the local governmental authorities under the reference number 55.2-1-54-2531-84-09, Regierung von Oberbayern.

### 2.6. Cell Adhesion Assay under Flow Conditions

Analysis of the THP-1 cell adhesion to a HUVEC monolayer under flow conditions was performed using an ibidi pump system (ibidi, Gräfelfing, Germany). Briefly, HUVECs (1.6 × 10^6^ cells/mL) were seeded into collagen G pre-coated μ-slide I^0.8^ Luer channel slides (ibidi, Germany), let to attach for 2 h and subjected to 1 h of 1 dyn/cm^2^, 1 h of 3 dyn/cm^2^ and constant 5 dyn/cm^2^ shear pressure overnight under cell culture conditions. After overnight cultivation and subsequent treatment as indicated in the respective figure legend, the monolayers were tested for the adhesion of THP-1 cells (8 x 10^5^ cells/mL), which were stained with 5 µM CellTracker Green (CTG; Life Technologies, Carlsbad, CA, USA) in serum-free RPMI for 30 min according to the manufacturer’s instructions. Adhesion was performed for 10 min at 0.5 dyn/cm^2^ under cell culture conditions. Non-adherent THP-1 cells were removed by washing once with PBS containing Ca^2+^ and Mg^2+^ and the cells were lysed using RIPA and frozen at −80 °C. Quantification of the THP-1 cell adhesion was performed by fluorescence measurement (ex.: 485 nm, em.: 535 nm) of the obtained lysates by using an Infinite F200 pro microplate reader (Tecan Trading, Männedorf, Switzerland).

### 2.7. Flow Cytometric Analysis

To analyze the cell surface levels of CAMs, HUVECs were washed twice with PBS and detached with HyQTase (GE Healthcare, Solingen, Germany). Detached HUVECs were stained on ice with either FITC-labeled anti-human CD54 (ICAM-1) antibody (MCA1615F, Biozol, Eching, Germany), PE-labeled anti-human CD106 (VCAM-1) antibody (555647, Becton Dickinson, Heidelberg, Germany) or PE-labeled anti-human CD62 antigen-like family member E (E-selectin) antibody (551145, Becton Dickinson, Germany) in PBS containing 0.2% BSA for 45 min. Unbound antibody was removed by washing twice with PBS containing 0.2% BSA. The protein levels on the endothelial cell surface were analyzed using flow cytometry (FACSVerse, Becton Dickinson, Germany).

### 2.8. Western Blot Analysis

HUVECs were washed once with PBS and lysed with RIPA lysis buffer containing Complete Mini (Roche, Mannheim, Germany), phenylmethylsulfonyl fluoride (PMSF), sodium fluoride (NaF) and sodium orthovanadate. For detection of phosphorylation sites, the lysis solution was additionally supplemented with β-glycerophosphate, sodium pyrophosphate and H_2_O_2_. Subsequently, lysates were transferred to TPX-tubes (Diagenode, Seraing, Belgium) and sonicated at ‘high’ setting in a Bioruptor plus system (Diagenode, Belgium) for 2 cycles with 30 s on- and 30 s off-intervals. The total protein concentration was determined using the Pierce BCA Protein Assay Kit (Thermo Fisher Scientific, Dreieich, Germany) according to the manufacturer’s instructions. Lysates were supplemented with reducing SDS-PAGE sample buffer and proteins were separated by discontinuous SDS-PAGE (Biorad, Feldkirchen, Germany) and transferred to a polyvinylidene fluoride membrane by tank electroblotting (Biorad, Germany). Membranes were blocked for unspecific protein binding with 5% non-fat dry milk or BSA in TBS-T and were incubated with primary antibody. Anti-human ICAM-1 polyclonal rabbit antibody (#4915), phospho-IKKα/β rabbit monoclonal antibody (#2697), IKKβ rabbit monoclonal antibody (#8943), phospho-IκBα rabbit monoclonal antibody (#2859), IκBα rabbit monoclonal antibody (#4812), phospho-SAPK/JNK rabbit monoclonal antibody (#4668) and SAPK/JNK rabbit polyclonal antibody (#9252) were obtained from Cell Signaling Technologies (Leiden, Netherlands). Anti-human VCAM-1 monoclonal mouse antibody (sc-13705), E-selectin mouse monoclonal antibody (sc-137054), NFκB-p65 mouse monoclonal antibody (sc-8008), AP-1-cJun mouse monoclonal antibody (sc-74543) and Brd4 rabbit polyclonal antibody (sc-48772) were obtained from SantaCruz Biotechnologies (Heidelberg, Germany). Horse radish peroxidase (HRP)-conjugated secondary anti-rabbit (#7074) or anti-mouse (#7076) antibodies were obtained from Cell Signaling (Netherlands) and used for visualization. Densitometric analysis was performed using the program ImageJ.

### 2.9. Western Blot Analysis of Cell Fractions

For preparation of cell fractions prior to western blot analysis, HUVECs were grown to confluency in 10 cm dishes. After treatment, the cells were washed once with PBS, scraped off and resuspended in hypotonic buffer (20 mM Tris-HCl pH 7.4, 10 mM KCl, 2 mM MgCl_2_, 1 mM EGTA) supplemented with protease and phosphatase inhibitors (0.5 mM dithiothreitol (DTT), PMSF, NaF, sodium orthovanadate, β-glycerophosphate, sodium pyrophosphate and Complete Mini) and let to swell on ice for 15 min. Subsequently, 0.75% (*v*/*v*) Nonidet P40 (NP-40) was added and the cells were vortexed at max setting for 10 s. To separate the cytoplasmic lysate, nuclei were sedimented at 17,000 *g* for 1 min at 4 °C and the supernatant (cytoplasmic fraction) was subtracted. Nuclei were washed once by resuspension in isotonic buffer (20 mM Tris-HCl pH 7.4, 150 mM KCl, 2 mM MgCl_2_, 1 mM EGTA and 0.3% (*v*/*v*) NP40) supplemented with protease and phosphatase inhibitors and incubated on ice for 5 min. Subsequently, the nuclei were sedimented for 3 min at 1,000 g and 4 °C and resuspended in RIPA supplemented with protease and phosphatase inhibitors. For lysis, the nuclei were sonicated in TPX-tubes (Diagenode, Belgium) at the ‘high’ setting in a Bioruptor plus system (Diagenode, Belgium) for two times with two cycles at 30 s on- and 30 s off-intervals. The total protein content of the cytoplasmic and nuclear fractions was determined as described above and samples were supplemented with reducing SDS-PAGE sample buffer.

### 2.10. Quantitative Polymerase Chain Reaction (qPCR)

The total RNA of treated HUVECs was isolated using the ion exchange chromatography-based RNeasy Mini Kit (Qiagen, Hilden, Germany) with additional on-column DNA digestion using the RNase-free DNase set (Qiagen, Germany) according to the manufacturer’s instructions. Following isolation, 1 µg of total RNA was transcribed into cDNA using the SuperScript II Reverse Transcriptase (Invitrogen, Waltham, MA, USA). qPCR was performed based on the 2^−ΔΔCt^ method (Livak KJ et al. [32]) with the StepOnePlus real-time PCR system (Applied Biosystems, Foster City, CA, USA) using the Power SYBR Green PCR Master Mix (Life Technologies, USA). The *gapdh* levels were used for normalization. All of the primers used for qPCR analysis of the mRNA levels are listed in the Appendix A.

### 2.11. mRNA Decay

To test the decay rate of mRNA, HUVECs were pre-treated with TNF (10 ng/mL) for the indicated times and supplemented with 2 µg/mL actinomycin D. Subsequently, the respective MTA was added, and samples were taken at different treatment times. Isolation of RNA and subsequent qPCR analysis was performed as described above. Results were normalized to the *gapdh* content of the samples treated with TNF only.

### 2.12. Reporter Gene Assay

The reporter gene assay was performed based on the Dual-Luciferase reporter assay system (Promega, Walldorf, Germany) according to the manufacturer’s instruction. Cells were transfected with plasmids bearing either the NFκB response element (pGL4.32[*luc2p*/NF-κB-RE/Hygro], Promega, Germany) or the AP-1 response element (pGL4.44[*luc2p*/AP-1-RE/Hygro], Promega, Germany) with the downstream *Photinus* luciferase reporter gene, together with a constitutively expressed *Renilla* luciferase-based control vector (pGL4.74[*hRluc*/TK]) for normalization purposes, using the HUVEC Nucleofactor Kit (Lonza, Basel, Switzerland) with the electroporator Nucleofactor 2b (program: A034; Lonza, Switzerland) according to the manufacturer’s instructions. Reporter gene luminescence activity was analyzed (*Photinus* luciferase em.: 560 nm/*Renilla* luciferase em.: 480 nm) using an Infinite Pro F200 plate reader (Tecan Trading, Switzerland) with the Te-Inject reagent injector (Tecan Trading, Switzerland).

### 2.13. Chromatin Immunoprecipitation (ChIP) qPCR

HUVECs were grown to confluency in 10 cm dishes and treated as indicated. After treatment, the cells were washed once with PBS, and protein complexes were cross-linked using 2 mM disuccinimidyl-glutarate (DSG; SantaCruz Biotechnologies, Germany) in PBS for 45 min at room temperature while shaking. Subsequently, the cells were washed three times with PBS, and proteins were cross-linked to the DNA by adding 1% (*v*/*v*) methanol-free formaldehyde (Polysciences, Hirschberg an der Bergstraße, Germany) in PBS for 10 min at room temperature. The cross-linking reaction was quenched by addition of 125 mM glycine for 5 min at room temperature. The cross-linked cells were washed twice with ice-cold PBS, scraped off and sedimented at 1,000 g and 4 °C for 5 min and resuspended in 375 µL of ChIP lysis buffer (50 mM HEPES-KOH pH 7.5, 140 mM NaCl, 1 mM EDTA, 1% (*v*/*v*) Triton X-100, 1% (*w*/*v*) SDS and 0.1% (*w*/*v*) Sodium deoxycholate) supplemented with PMSF and Complete Mini and incubated on ice for 10 min while shaking. Subsequently, 187.5 µL of lysates were transferred into 1.5 mL TPX-tubes (Diagenode, Belgium), and the chromatin was sheared at the ‘high’ setting with a Bioruptor plus system (Diagenode, Belgium) at 4 °C for three times with 10 cycles at 30 s on- and 30 s off-intervals to obtain chromatin fragments of 200–600 bp length. After shearing, cell debris was sedimented at 8000 *g* for 10 min at 4 °C and 100 µL of the obtained supernatant was diluted 1:10 in RIPA supplemented with PMSF and Complete Mini. 1 µg of the respective antigen binding or negative control antibody was added, and samples were incubated overnight at 4 °C while rotating. Anti-human NFκB-p65 rabbit polyclonal antibody (C15310256), anti-human RNA polymerase II mouse monoclonal antibody (C15200004), anti-human Brd4 rabbit polyclonal antibody (C15410337), negative control mouse IgG antibody (C15400001) and negative control rabbit IgG antibody (C15410206) was obtained from Diagenode (Belgium), anti-human AP-1-cJun polyclonal rabbit antibody (24909-1-AP) was obtained from Proteintech (Planegg-Martinsried, Germany) and anti-human tri-methyl-histone H3 monoclonal rabbit antibody was obtained from Cell Signaling Technologies (Netherlands). To immunoprecipitate the antibody-chromatin complexes, 20 µL of Protein A/G magnetic beads (88802; Thermo Fisher, Germany), pre-blocked with herring sperm DNA and BSA, were added to each sample and incubated for 4 h at 4 °C while rotating. Subsequently, the beads were washed once with low salt washing buffer (20 mM Tris-HCL pH 8.0, 150 mM NaCl, 2 mM EDTA, 1% (*v*/*v*) Triton X-100 and 0.1% (*w*/*v*) SDS), high salt washing buffer (20 mM Tris-HCL pH 8.0, 500 mM NaCl, 2 mM EDTA, 1% (*v*/*v*) Triton X-100 and 0.1% (*w*/*v*) SDS) and lithium chloride-containing washing buffer (10 mM Tris-HCL pH 8.0, 250 mM LiCl, 1 mM EDTA, 1% (*v*/*v*) NP-40 and 1% (*w*/*v*) sodium deoxycholate) and eluted in 120 µL elution buffer (100 mM NaHCO_3_ and 1%(*w*/*v*) SDS) at 30 °C for 15 min while shaking. After removal of the beads, 4.8 µL of 5 M NaCl and 2 µL of 10 mg/mL RNase A (10109142001; Roche, Germany) was added and incubated overnight at 65 °C. The next day, 2 µL of 20 mg/mL proteinase K was added and incubated for 1 h at 60 °C. In case of percent-input ChIP, 50 µL of the initially sheared chromatin lysate were supplemented with 70 µL of elution buffer and subjected to the same treatment as described above. The DNA was isolated using the ChIP DNA Clean and Concentrator Kit (Zymo Research, Freiburg, Germany) according to the manufacturer’s instructions and used for qPCR analysis. qPCR was performed using ChIP primers encompassing the most proximal transcription factor binding sites in the promoters of target genes (p65, cJun) or non-translated regions downstream of the transcription start site where RNA polymerase stalling occurs (RNA polymerase II and Brd4). For analysis of transcription factor and RNA polymerase enrichment, data was normalized on the background levels of a sequence of the *rpl13a* 3′-untranslated region. Brd4 enrichment data was analyzed based on the percent-input method using the equation:$\;100\times 2^{\left({Ct}_{input}-{Ct}_{IP}\right)}$. All of the primers used for ChIP-qPCR analysis are listed in the Appendix A.

### 2.14. Protein Translation Inhibition Assay

HUVECs were seeded into 8-well µ-ibidi slides (ibidi, Germany), grown to confluence and treated as indicated. The cells were fixated with 4% Histofix (Roth, Karlsruhe, Germany) formaldehyde solution and were permeabilized using 0.2% Triton X-100 (Merck, Darmstadt, Germany) in PBS. Unspecific binding was blocked using 0.2% BSA in PBS. Protein translation was assayed using the Click-IT Plus OPP Protein Synthesis Assay Kit (Thermo Fisher, Germany) according to the manufacturer’s instructions with a chemoselective Alexa488 picolyl azide dye to visualize de novo protein synthesis and a HCS NuclearMask blue dye (1:2000) in PBS for nuclei staining. The AlexaFluor488 fluorescence was normalized to the nuclei fluorescence. Imaging was performed using a Leica DMI6000 B fluorescence microscope (Leica Microsystems, Mannheim, Germany). Densitometric image analyzation of the nuclei was performed using the software ImageJ.

### 2.15. Metabolic Activity Assay

HUVECs were seeded into a 96-well plate and were grown to confluence and treated as indicated. The metabolic activity was tested with the CellTiterBlue Cell Viability Assay (Promega, Germany) following the manufacturer’s instructions. The metabolism-dependent reduction of resazurin into resorufin was analyzed by fluorescence measurement (ex.: 579 nm, em.: 584 nm) using an Infinite F200 pro microplate reader (Tecan Trading, Switzerland).

### 2.16. Lactate Dehydrogenase (LDH) Release Assay

Release of LDH from treated HUVECs was measured with the CytoTox96 non-radioactive cytotoxicity assay (Promega, Germany). Based on the manufacturer’s instructions, the absorbance of a red formazan product (480 nm) was measured resulting from the reduction of iodonitrotetrazolium salt by NADH. Absorbance measurements were performed using a Varioskan Flash microplate reader (Thermo Fisher Scientific, Germany).

### 2.17. Cell Viability Assay

HUVECs were seeded into 24-well plates, grown to confluence and treated with the respective MTA or staurosporine as indicated. Cell viability was judged by measuring sub-diploid cells based on the method developed by Nicoletti et al. [33]. Briefly, treated cells were detached with trypsin/EDTA (PanBiotech, Germany) in H_2_O and were stained overnight at 4 °C in PBS containing 0.1% Triton X-100, 0.1% sodium citrate and 50 µg/mL propidium iodide. Analysis was performed by flow cytometry (FACSVerse, Becton Dickinson, Germany).

### 2.18. siRNA Transfection

Transfection of HUVECs was performed with the GeneTrans II transfection reagent (MoBiTec, Goettingen, Germany) based on the manufacturer’s instructions. Briefly, HUVECs were seeded into 6-well plates (25,000 cells/cm^2^) and cultivated overnight. Transfection was performed at a confluence state of 70–80% with either smart-pool non-targeting siRNA (D-001810-10-05) or smart-pool siRNA targeting Brd4 (L-004937-00-0005) purchased from Dharmacon (Lafayette, CO, USA) in FCS-free ECGM (PeloBiotech, Germany) supplemented with 2 mM of L-glutamine (Thermo Fisher Scientific, Germany) under cell culture conditions. After 4 h, the transfected cells were supplied with fresh ECGM supplemented with 10% (*v*/*v*) FCS (PanBiotech, Germany) and were incubated overnight. On the next day, transfected cells were again supplied with fresh medium. Experiments were performed 48 h after the initial transfection.

### 2.19. Statistical Analysis

Data analysis was performed with the software GraphPad Prism 7.0 (GraphPad Software, Boston, MA, USA). The One-way ANOVA and Tukey’s post-hoc test was used for evaluation of data resulting from three or more independently performed experiments. The numbers of performed replicates (n) are shown in the figure legends. Results were considered as statistically significant when *p* ≤ 0.05.

## 3. Results

### 3.1. Pretubulysin Reduces Inflammation In Vivo in a Psoriasiform Dermatitis Mouse Model

To initially assess the potential of pretubulysin (PT) in treating inflammation *in vivo*, we tested the effects of PT on the inflammatory response in a murine imiquimod (IMQ)-induced psoriasiform dermatitis model. The IMQ-induced inflammatory response and resulting psoriasiform dermatitis peaked 4 days after IMQ application, as based on the increased redness, thickness, scaling and the resulting cumulative Psoriasis Area Severity Index (PASI) score. Treatment with PT (1 mg/kg) significantly reduced the redness of the skin by approx. 50% and thickness of the skin by approx. 30% after 4 days of IMQ-treatment, while not influencing the skin scaling (Figure 1). This led to a reduction of the resulting PASI score by approx. 33%, compared to the control animals. These results clearly demonstrate that PT reduces the inflammatory response *in vivo*.

### 3.2. Pretubulysin Reduces TNF-Induced Inflammatory Processes In Vivo and In Vitro by Down-Regulation of Endothelial Components of the Leukocyte Adhesion Cascade

To further assess if the anti-inflammatory effects observed for PT are related to an inhibition of the leukocyte adhesion cascade, we tested the influence of PT treatment on the interaction of leukocytes with the vascular endothelium via intravital microscopy of the TNF-activated mouse cremaster muscle. The crucial steps of the leucocyte extravasation cascade, i.e., rolling, firm adhesion and transmigration, were analyzed. Treatment with PT (1 mg/kg) caused a strong reduction in the firm adhesion (approx. 75%) and transmigration (approx. 65%) of the leukocytes onto and through the TNF-activated endothelium (Figure 2a, mid and right). The rolling flux of leukocytes on the endothelial surface was marginally, but not significantly increased (approx. 25%) (Figure 2a, left). To confirm the inhibitory action of PT on the interaction of leukocyte with endothelial cells in vitro, we analyzed the effects of PT on the firm adhesion of cells of the human monocytic cell line THP-1 onto PT-treated and TNF-activated HUVECs under flow conditions (Figure 2b). Of note, only the endothelium was treated in this assay. Treatment with PT (300 nM) significantly reduced the TNF-triggered firm adhesion of THP-1 cells to the HUVEC monolayer by approx. 65% compared to the TNF control cells.

To ensure that the observed effects are not based on cytotoxicity, we analyzed PT and the therapeutically used microtubule-destabilizers colchicine (COL) and vincristine (VIN), as well as the microtubule-stabilizer paclitaxel (PAC) in cell viability assays. Neither of the compounds tested had any significant negative impact on the metabolic activity (Appendix A), membrane integrity (Appendix A) or apoptosis rate (Appendix A) of HUVECs at any of the concentrations tested (10, 30, 100, 300, 1000 nM). For PT, only the highest concentration tested (1000 nM) induced a slight, but not significant increase in the rate of apoptosis (Appendix A, top left). To exclude negative effects of the compounds on protein translation, we analyzed the overall de novo protein synthesis in HUVECs after treatment with the MTAs (Appendix A). Treatment with PT induced a time-dependent increase in the de novo protein synthesis compared to the control cells (Appendix A), with TNF inducing no additional effects concerning the protein translation (Appendix A). Notably, only the destabilizing MTAs induced an apparent increase in the de novo protein synthesis, while the stabilizer PAC showed no effect (Appendix A). Previous studies investigating the effects of destabilizing MTAs on the leukocyte adhesion cascade implied an MTA-induced down-regulation of the TNF-receptor 1 (TNFR1) levels and a resulting desensitization of cells to TNF. Therefore, we tested the total protein levels of TNFR1 by western blot analysis (Appendix A) and found that neither short (0.5 h) nor long (24 h) treatment periods with PT reduced the total TNFR1 levels in HUVECs.

To analyze if the inhibitory effects of PT concerning the leukocyte adhesion are related to a down-regulation of the CAMs, we tested the effects of PT on the levels of ICAM-1 and VCAM-1 on the cell surface in HUVECs 24 h after TNF activation (Figure 2c). While low concentrations (10 nM) of PT had no apparent effect on the TNF-evoked rise in ICAM-1 cell surface levels, the TNF-induced VCAM-1 levels were slightly, but significantly increased. Higher concentrations of PT (30, 100, 300, 1000 nM) led to a significant and concentration-dependent reduction of the cell surface levels of these CAMs, with ICAM-1 levels being reduced to approx. 35% at 300 nM and 15% at 1000 nM of PT, while VCAM-1 levels were reduced to approx. 10% at 300 nM and 5% at 1000 nM of PT compared to the TNF-activated cells.

Regarding E-selectin (Appendix A), which is involved in capture and rolling of leukocytes, we found that PT at 10 and 30 nM marginally increased the TNF-induced E-selectin levels, while it concentration-dependently decreased the levels at 100 to 1000 nM. PT only slightly reduced the E-selectin levels by approx. 10% at a concentration of 300 nM and by approx. 25% at a concentration of 1000 nM.

Flow cytometric analysis of the ICAM-1 and VCAM-1 cell surface levels after treatment with VIN and COL (300 nM) revealed a reduction of the CAM levels similar to that of PT (Figure 3a, top and bottom), with VIN and COL also inducing a concentration-dependent inhibition of the CAM surface levels (Appendix A). In contrast, treatment with 300 nM of the microtubule-stabilizer PAC increased the CAM levels by approx. 30% compared to the TNF control (Figure 3a, top and bottom). The induction of the CAM levels over the TNF control could already be observed after treatment with 100 nM of PAC (Appendix A) and was also present after treatment with 1000 nM of the microtubule-stabilizer.

Due to the intensity of the observed down-regulation of the CAMs important for the firm adhesion of leukocytes on the endothelial cell surface after treatment with 300 nM of PT (and the other destabilizing MTAs) and the potential negative effects on the cell viability at higher concentration (1000 nM), we decided to carry out further testing of PT at a concentration of 300 nM.

Since the microtubule-destabilizing action of MTAs could affect the secretory pathways, we analyzed the effect of PT on the total protein and relative mRNA levels of ICAM-1 and VCAM-1 by western blot and quantitative PCR (qPCR) analysis, respectively. While treatment with PT reduced the total protein and mRNA levels of ICAM-1 by approx. 75% compared to the TNF control (Figure 3b,c, top), VCAM-1 total protein and relative mRNA levels were reduced to baseline values (Figure 3b,c, bottom) indicating almost full inhibition of the TNF-induced VCAM-1 synthesis. VIN and COL showed nearly identical effects on the CAM total protein and relative mRNA levels (Appendix A). However, treatment with PAC slightly increased the total protein levels and induced a weak reduction of the relative mRNA levels of both CAMs. Surprisingly, all tested MTAs decreased the TNF-induced total protein levels of E-selectin (approx. 60%; Appendix A) to a similar extend as seen for ICAM-1, but only the destabilizers showed a comparable reduction of the relative *e-selectin* mRNA levels (PT: approx. 60%, VIN/COL: approx. 50%; Appendix A), while PAC only had minor effects (approx. 10%).

Taken together, these results show that the microtubule destabilizer PT (as well as VIN and COL) decreases the interaction of leukocytes with the endothelium by down-regulation of the leukocyte adhesion components ICAM-1 and VCAM-1 in HUVECs and hence reduces the firm adhesion of leukocytes, as well as the subsequent transmigration trough the endothelial monolayer, while the microtubule stabilizer PAC shows no apparent inhibitory potential concerning the components of the leukocyte adhesion cascade.

### 3.3. Pretubulysin Increases the TNF-Induced NFκB and AP-1 Signaling Cascades in HUVECs, but Reduces the Promoter Activity of NFκB and AP-1

All of the CAMs that are involved in the leukocyte adhesion cascade share transcription factor motifs for NFκB and AP-1 in their promoters. Thus, we tested the influence of PT on the pro-inflammatory NFκB and AP-1 signaling cascades by western blot analysis and reporter gene assay in TNF-activated HUVECs (Figure 4 and Figure 5). Treatment with PT increased the phosphorylation ratio and hence the activity of the IκB kinase (IKK) compared to the TNF-treated control cells at all tested treatment times (Figure 4a). Notably, while PT only slightly increased IKK phosphorylation after short TNF stimulation times (0.5 h), high IKK activity prevailed at longer TNF stimulation times in the PT-treated cells, when the TNF-induced IKK activity already diminished (2, 6, 16 h). In addition, extended TNF stimulation also led to a small but noticeable decrease in the total IKK protein levels independent of PT treatment.

In consequence, we tested the effects of PT on the phosphorylation and total protein levels of IκBα (Figure 4b). TNF increased the IKK-dependent phosphorylation and strongly reduced the total protein levels of IκBα after 0.5 h, which increased again after longer stimulation times. In comparison, PT induced an even stronger degradation of IκBα at 0.5, 2 and 16 h of TNF stimulation, but appeared to slightly induce IκBα levels after 6 h of stimulation. Therefore, we tested the IκBα protein levels at mean treatment times of 10 h and 12 h (Appendix A) and found that PT increased the IκBα protein levels by approx. 55% after 10 h and by 25% after 12 h of TNF stimulation compared to the TNF control. Thus, it is likely that PT induces a time-dependent differential expressional pattern of IκBα, which might negatively influence the NFκB response at these treatment times.

Subsequently, we also tested the NFκB-p65 protein levels in the cytoplasm and the nucleus of TNF-activated HUVECs after PT treatment by western blot analysis of cell fractions (Figure 4c; western blot analysis of the purity of fractions is shown in Appendix A). 2 h of TNF stimulation induced a strong and lasting translocation of p65 into the nucleus, which only weakly diminished after 16 h. In comparison, treatment with PT significantly reduced the p65 levels in both fractions after 16 h, hinting at an increased degradation. In addition, we also tested the relative *p65* mRNA levels after TNF stimulation and PT treatment (Figure 4d). A short TNF stimulation of 0.5 h had no effect on the relative *p65* mRNA levels, but extended stimulation of 2 and 6 h led to a time-dependent increase of *p65* mRNA, with PT further inducing the relative *p65* mRNA levels compared to the TNF controls. To evaluate the influence of PT on the transcriptional activity of NFκB, we performed a reporter gene assay after 6 h of TNF stimulation (Figure 4e), when the apparent nuclear NFκB-p65 levels are not influenced by PT. Treatment with 100-1000 nM of PT led to a significant reduction of the TNF-induced promoter activity of NFκB by approx. 60%. This apparent decrease in activity of NFκB at times when the relative NFκB-p65 levels remain unchanged indicates a reduced DNA-binding activity or transactivatory potential of this transcription factor. The study of Schwabe et al. [34] has correlated an increased IKK activity with a decreased NFκB-p65 transactivatory potential at the promoters of *icam-1* and *vcam-1* via an increased phosphorylation of NFκB-p65 at serine 468. In addition, further studies have implied the NFκB-p65 phosphorylation at serine 536 to negatively influence the stability of NFκB-p65 (Pradère et al. [35]). Since IKK can phosphorylate NFκB-p65 at both sites and since PT treatment increased IKK activity, we tested the influence of PT on the levels of both NFκB phosphorylation sites by western blot analysis. We found that PT had no significant influence on the phosphorylation levels of p65-S468 and p65-S536 compared to the TNF control (Appendix A).

Due to the dimeric nature of NFκB and the increased DNA-binding affinity of the heterodimeric p65/p50 complex compared to the p65/p65 homodimer, we also tested the p50 protein and relative mRNA levels after PT-treatment (Appendix A). Compared to the TNF control, the total p50 protein levels in PT-treated HUVECs remained unchanged after 0.5 and 2 h, but were reduced by approx. 60% after 6 h of TNF stimulation (Appendix A). Analysis of the corresponding *p105* mRNA levels (Appendix A) revealed that TNF induced the mRNA levels of the p50 precursor only after 2 h of stimulation. Treatment with PT moderately increased the relative *p105* mRNA levels after 2 h, but slightly decreased the mRNA levels after 6 h of stimulation compared to the TNF controls.

To evaluate the impact of PT on the TNF-induced AP-1 signaling cascade, we first tested the effects of PT-treatment on the activity of the AP-1-activating upstream kinase JNK in TNF-activated HUVECs (Figure 5a). TNF stimulation increased the phosphorylation of JNK after 0.5 h by approx. 4-fold compared to control cells, which subsequently increased to approx. 5-fold after 2 h and 10-fold after 6 h. In contrast, treatment with PT increased the phosphorylation of JNK approx. 30- to 40-fold compared to control cells at all tested treatment times, with highly phosphorylated JNK being present in the cytoplasm as well as the nucleus of HUVECs (Figure 5c). cJun constitutes the main component of the transcription factor complex AP-1. While JNK induces the transactivatory potential of cJun by phosphorylation, high and extended JNK activity is known to correlate with reduced cJun DNA binding and increased cJun degradation. However, we found that PT increased the cJun protein levels compared to the TNF controls after all stimulation times tested (Figure 5b). Comparison with the control cells revealed that stimulation with TNF for 0.5 h induced an apparent and phosphorylation-dependent electromobility shift of cJun, which was further increased by PT. Importantly, this electromobility shift is also present after 16 h of TNF stimulation, at a time when it is almost fully lost in the TNF control cells, indicating the induction of high and lasting cJun activity due to PT. Since cJun is known to auto-induce itself by binding to the *cjun* promoter, we also tested the relative *cjun* mRNA levels in dependency of PT treatment (Figure 5d). In accordance with the elevated cJun protein levels, PT treatment led to increased *cjun* mRNA levels (approx. 50%) after 0.5 h of TNF stimulation. Longer stimulation periods did not lead to an induction of *cjun* mRNA levels higher than the negative control cells under neither of the treatment conditions tested. Again, we also tested the transcriptional activity of AP-1 after PT treatment in a reporter gene assay after 6 h of TNF stimulation, when the apparent cJun protein levels are not influenced by PT (Figure 5e). PT significantly reduced the TNF-activated AP-1 promoter activity in HUVECs, with concentrations of 100 nM or more leading to a reduction of the promoter activity of AP-1 by approx. 50%. In addition to PT, we also tested the NFκB and AP-1 promoter activity after treatment with VIN, COL and PAC (Appendix A). While VIN decreased the TNF-induced promoter activity of NFκB and AP-1 to a similar extend than PT, the effects of COL were slightly weaker. Surprisingly, the microtubule stabilizer PAC also decreased the NFκB and AP-1 promoter activity, albeit to a lesser extend then COL.

In general, the increased activity of IKK and JNK after treatment with PT shows that PT induces the pro-inflammatory signaling of these NFκB and AP-1 upstream kinases. Especially in case of the AP-1-cJun signaling cascade, the high activity of JNK led to increased cJun protein levels and likely cJun activity, as based on the phosphorylation-dependent electromobility shift present at all treatment times. In case of the NFκB signaling cascade, the elevated IKK activity appeared to increase the degradation of IκBα. However, while the IKK-dependent phosphorylation of NFκB-p65, as well as the nuclear p65 levels remained largely unchanged, extended treatment with PT led to reduced p65 levels in the cytoplasm and nucleus of HUVECs. The fact that PT significantly reduced the promoter activity of NFκB and AP-1, at the same times when the protein levels of both transcription factors were largely uninfluenced by PT is of particular interest and might indicate a reduced DNA-binding affinity of NFκB and AP-1 after treatment with PT.

### 3.4. The Reduced Promoter Activity of NFκB-p65 and AP-1-cJun Is Not Due to a Decreased DNA-Binding Activity These Transcription Factors

To explore the possibility of reduced DNA-binding of the NFκB-p65 and AP-1-cJun transcription factors to the promoters of *icam-1* and *vcam-1*, we performed chromatin immunoprecipitation (ChIP; Figure 6). PT treatment increased the p65 enrichment following 2 h of TNF stimulation in the *icam-1* (Figure 6a) promoter by an approx. 75% and in the *vcam-1* (Figure 6b) promoter by an approx. 40%. Longer TNF stimulation of 6 h led to a minor decrease in p65 enrichment for *icam-1*, but decreased the p65 enrichment for *vcam*-1 to the value of the TNF control. Extended stimulation of 16 h further reduced the p65 enrichment in the *icam-1* promoter of the PT-treated cells to a value similar to that of the TNF control and further reduced the p65 enrichment in the *vcam-1* promoter to approx. 60% of the TNF control. In case of AP-1-cJun, enrichment in the *icam-1* (Figure 6c) and *vcam-1* (Figure 6d) promoter was similarly increased after 6 h of TNF stimulation, to approx. 270% in the *icam-1* promoter and 160% in the *vcam-1* promoter. Testing the cJun enrichment at shorter treatment times (2 h, Appendix A), also showed increased cJun enrichment in the promoters of both CAM genes. In addition to *icam-1* and *vcam-1*, we also tested the p65 and cJun enrichment in the promoter of *e-selectin* (Appendix A). While PT-treatment led to a very similar time-dependent pattern of p65 enrichment to that of *icam-1*, cJun enrichment in the promoter of *e-selectin* was not significantly influenced after 2 h of TNF stimulation, but strongly enriched after 6 h of TNF stimulation and PT-treatment.

Taken together, these results show that the PT-induced decrease in the promoter activity of NFκB and AP-1 after 6 h of TNF stimulation is not due to a decreased DNA-binding activity of the main NFκB and AP-1 transcription factor components p65 and cJun, as shown by ChIP.

### 3.5. Treatment with Pretubulysin Leads to a Time-Dependent Reduction of RNA Polymerase II Enrichment in the icam-1 and vcam-1 Genes Independent on cjun and p65 Enrichment and Promotes Super-Induction of the CAM mRNAs at Mean Pre-Treatment Times

Due to the time-dependent and differential enrichment of NFκB-p65 especially at longer TNF stimulation times, we further aimed at analyzing how the differences in transcription factor enrichment might influence the recruitment of the RNA polymerase II (Pol II) to the *icam-1* and *vcam-1* genes. We performed ChIP analyses with primers encompassing the *icam-1* and *vcam-1* promoter regions and the Pol II stalling regions, but found that enrichment for both regions was highly similar (Appendix A). PT-treatment moderately reduced the Pol II enrichment in the stalling regions compared to the TNF controls after 2 h of TNF stimulation to 60% for *icam-1* and 50% for *vcam*-1 (Figure 7a,b). Interestingly, following 6 h of TNF stimulation, the PT-induced decrease in Pol II enrichment for *icam-1* was fully lost, while the enrichment in the *vcam-1* stalling region increased to approx. 80% of the TNF control. Extended TNF stimulation of 16 h again reduced the TNF-induced Pol II enrichment in the *icam-1* and *vcam-1* stalling regions to an approx. 30% and 10% of the TNF control, respectively. Additionally, we also tested the Pol II enrichment after PT treatment in the *e-selectin* stalling region (Appendix A). The Pol II enrichment in the *e-selectin* gene was similarly reduced as seen for *vcam-1* after 2 h and 16 h of TNF stimulation, though approached the same levels of enrichment than observed for *icam-1* after 6 h of TNF stimulation. Based on the time-dependent enrichment of Pol II in both CAM genes, we subsequently tested if and how different PT pre-treatment times might influence the relative mRNA levels of ICAM-1 and VCAM-1 (Figure 7c). Stimulation with TNF for 0.5 h increased relative *icam-1* and *vcam-1* mRNA levels approx. 10-fold compared with control cells. Pre-treatment with PT for short (0.5 h to 2 h) and long (16 h to 24 h) durations led to the expected decrease of the relative CAM mRNA levels. In contrast, mean pre-treatment times of 4 h to 8 h significantly increased the relative CAM mRNA levels compared to the TNF control. For both CAMs, mRNA levels peaked after 6 h of pre-treatment, with *icam-1* mRNA levels being increased approx. 40-fold and the *vcam-1* mRNA levels being increased approx. 100-fold compared to the control cells. In case of *e-selectin*, mRNA levels followed a similar time-dependent induction of mRNA (Appendix A). However, the initial induction of the *e-selectin* mRNA after 0.5 h of TNF stimulation was significantly higher, with a 300-fold increase of the relative mRNA levels compared to the control cells. In addition, *e-selectin* mRNA levels after 6 h of PT pre-treatment did not exceed the mRNA levels of the TNF control.

Based on the differences observed for the enrichment of NFκB-p65 and AP-1-cJun compared to the transcription factor-dependent enrichment of Pol II at and behind the promoters of *icam-1* and *vcam-1*, it is likely that PT exerts its effect largely independent of the transcription factor activity of NFκB-p65 and AP-1-cJun. In addition, the time-dependent enrichment of the RNA polymerase II and the super-induction of CAM mRNAs at treatment times, when the PT-influenced enrichment of the RNA polymerase II approaches TNF control levels, suggests a time-dependent and dual mechanism of the PT-induced down-regulation of the CAMs.

### 3.6. The Pretubulysin-Evoked Down-Regulation of the Cell Adhesion Molecules ICAM-1 and VCAM-1 after Long-Term but Not Short-Term Treatment Can Be Reversed by Inhibition of JNK

Due to the apparent time-dependent differences of Pol II enrichment and the high and long-lasting JNK activity after PT treatment, we tested how inhibition of JNK might influence the relative mRNA levels at short-term and long-term treatment with PT and stimulation with TNF and how this inhibition might affect the cell surface levels of ICAM-1 and VCAM-1 (Figure 8). While TNF induction of the relative mRNA levels for 0.5 h only induced a 5-fold increase of *icam-1* (Figure 8a, left) and *vcam-1* (Figure 8b, left) mRNA, the relative mRNA levels of both CAMs were induced approx. 200-fold and 1500-fold after 2 h of TNF stimulation, respectively, and did not decrease below these values after 6 h or 16 h of TNF stimulation. PT treatment strongly reduced the levels of both CAM mRNAs at all treatment times. Inhibition of JNK by the irreversible JNK inhibitor JNKIN8 and TNF stimulation for 16 h decreased the relative *icam-1* mRNA levels to approx. 60%, but recovered the relative mRNA levels in the PT-treated cells from approx. 30% to 55% (Figure 8a, left and mid), suggesting a nearly full recovery of the PT-induced effects on relative *icam-1* mRNA levels. For *icam-1*, JNK inhibition slightly increased the mRNA levels compared to the TNF control and recovered the relative *vcam-1* mRNA levels for the PT-treated cells from approx. 2% to 30%. Conversely, while inhibition of JNK and stimulation with TNF for 1 h decreased the relative mRNA levels of both CAMs by approx. 50% (Figure 8a,b, right), JNK inhibition in the PT-treated HUVECs did not lead to recovery, but further decreased the relative mRNA levels. We also tested the PT- induced down-regulation of *e-selectin* mRNA levels for sensitivity to JNK inhibition which followed a similar pattern of recovery like observed for *icam-1* (Appendix A). To confirm the sensitivity of the PT-induced down-regulation of the CAMs on the activity of JNK, we tested how JNK inhibition influences the cell surface levels of ICAM-1 and VCAM-1 by flow-cytometry (Figure 8c). Inhibition of JNK and 24 h of TNF stimulation reduced the cell surface levels of ICAM-1 by approx. 50% and those of VCAM-1 by approx. 20%, indicating an increased sensitivity of the ICAM-1 promoter to JNK activity. Inhibition of JNK in PT-treated HUVECs recovered the ICAM-1 cell surface levels to the value of the JNKIN8-treated TNF control cells and recovered the VCAM-1 cell surface levels from 5% to 30%.

Taken together, these results show that the PT-induced down-regulation of the endothelial CAMs responsible for the adhesion (and rolling) of leukocytes after long-term, but not short-term PT treatment, can be partially recovered by inhibition of JNK, verifying the assumption that the PT-induced effects on the CAM levels follow a time-dependent dual mechanism of action. More importantly, the fact that the PT-induced down-regulation of *icam-1* (and *e-selectin*) can be fully recovered to what is possible under inhibition of JNK (because JNK activity contributes to CAM induction), suggests that the PT-induced JNK activity is the main cause for the inhibitory effects on these CAMs.

Considering the lack of negative regulation of the DNA-binding behavior of NFκB-p65 and AP-1-cJun in the promoters of the CAM genes after treatment with PT and the apparent dependency of the PT-induced effects on the activity of JNK, we hypothesize that PT exerts its effect downstream of the formation of the pre-initiation complex at the respective promoters of the target genes.

### 3.7. Pretubulysin Exerts Its Inhibitory Effects on the Cell Adhesion Molecule Synthesis in Part by Differentially Regulating the DNA-Binding Behavior and Protein Levels of the Histone-Acetyl Transferase Bromodomain-Containing Protein 4 via Induction of JNK

To elucidate how PT affects the synthesis of the CAMs downstream of the formation of the pre-initiation complex, we first tested the influence of PT-treatment on the protein levels and activity of the cyclin-dependent kinase 9 (CDK9). CDK9 is a component of the positive transcription elongation factor (pTEFb), which positively regulates the release of Pol II from its stalling site in the non-translated region of target genes succeeding the transcription start site. However, western blot analysis showed that the total CDK9 protein levels and the phosphorylation state were largely unaffected by both PT treatment and TNF stimulation (data not shown). In addition, we also tested if treatment with any of the MTAs might influence the decay rate of *vcam-1* mRNA (Appendix A). We found that treatment with the microtubule-destabilizers increased the half-life of *vcam-1* mRNA, while the microtubule-stabilizer PAC slightly reduced it.

In consequence, we investigated the influence of PT on the enrichment of the histone-acetyltransferase bromodomain-containing protein 4 (Brd4), which directly interacts with pTEFb and activates CDK9 via its kinase function. In addition, Brd4 has also been shown to directly interact with acetylated NFκB-p65 (p65-K310ac) via its bromodomain, hence having a particularly important function for the transcription of NFκB-dependent target genes. After 2 h of TNF stimulation, PT induced a significant increase in Brd4 enrichment in the Pol II stalling regions of *icam-1* by approx. 90% and *vcam-1* by approx. 50% (Figure 9a,b) compared to the TNF control cells. A longer TNF stimulation period and PT treatment time (6 h) led to a decrease in enrichment to approx. 95% in the *icam-1* stalling region and to 70% in the *vcam-1* stalling region. Extended TNF stimulation and PT treatment of 16 h led only to a minor further decrease of Brd4 enrichment for *icam-1*, but decreased the Brd4 enrichment in the *vcam-1* stalling region to approx. 50% compared to the TNF control. Regarding the histone-acetyltransferase function of Brd4, we further analyzed to what extend PT represses the induction of *icam-1* and *vcam-1* on the level of chromatin, by testing the enrichment of the repressive tri-methylation marks on histone 3-lysine 9 (H3K9me3) in the gene bodies of *icam-1* and *vcam-1* after 16 h of TNF stimulation (Figure 9c). ChIP analysis revealed that TNF induction for 16 h decreased the levels of H3K9me3 in the *icam-1* gene to an approx. 50% and *vcam-1* gene to approx. 40% compared to the control cells. Treatment with PT increased the enrichment of H3K9me3 to approx. 140% for *icam-1* and 125% for *vcam-1*, reflecting the observed inhibition of the CAM synthesis after PT treatment.

Due to the observed sensitivity of the PT-induced down-regulation of ICAM-1 and VCAM-1 on JNK inhibition, we tested how treatment of HUVECs with JNKIN8 influences the enrichment and hence DNA-binding behavior of Brd4 after PT-treatment (Figure 10a).

Inhibition of JNK increased the Brd4 enrichment following 6 h of TNF stimulation to approx. 150% for *icam-1* and 230% for *vcam-1*. For the PT-treated HUVECs, inhibition of JNK activity also significantly increased the enrichment of Brd4 for *icam-1* from approx. 85% to 115% and for *vcam-1* from approx. 75% to 150%. To further analyze the effects of PT on Brd4, we tested the nuclear Brd4 protein levels after treatment with PT and induction with TNF for different duration by western blot analysis (Figure 10b). Notably, Brd4 could only be detected in sufficient quantities in the nucleus of HUVECs. While TNF stimulation alone reduced the Brd4 protein levels over time, treatment with PT induced a stronger decrease in the Brd4 protein levels at all treatment times. Of note, the relative mRNA levels of Brd4 were not negatively influenced by PT treatment (Appendix A). In addition, inhibition of JNK reversed the TNF-induced reduction of the Brd4 protein levels and also slightly recovered Brd4 protein levels after PT-treatment (Figure 10c). To better understand how the PT-induced reduction in Brd4 protein levels and DNA-binding behavior influence the protein synthesis of ICAM-1 and VCAM-1, we transfected HUVECs with Brd4 small interfering RNA (Brd4si; Figure 10d). Western blot analysis revealed that transfection of HUVECs with Brd4si and incubation for 48 h significantly reduced the total protein levels of Brd4 to approx. 33% compared to the control cells transfected with non-targeting small interfering RNA (Figure 10d, left). Flow cytometric analysis of TNF-induced cell surface levels of ICAM-1 and VCAM-1 after 24 h of stimulation surprisingly revealed that the knock-down of Brd4 increased the cell surface levels of ICAM-1 by approx. 50%. PT-treatment further increased the levels of ICAM-1 on the cell surface of HUVECs (Figure 10d, middle). In the case of VCAM-1, knock-down of Brd4 strongly reduced the VCAM-1 protein levels after 24 h of TNF stimulation and only slightly led to a further reduction of the already significantly decreased VCAM-1 cell surface levels after PT-treatment (Figure 10d, right).

Taken together, these results show that the destabilizing microtubule-targeting agent PT partially exerted its effect on the protein synthesis of endothelial CAMs by negatively regulating the total Brd4 protein levels and differentially regulating the DNA-binding behavior of Brd4 in a JNK-dependent mechanism. At longer PT treatment times, this led to decreased levels of the histone-acetyltransferase Brd4 at the CAM genes, thereby enacting increased chromatin closure and repressing transcription. However, while knock-down of Brd4 led to the expected reduction of the TNF-induced cell surface levels of VCAM-1, ICAM-1 cell surface levels were increased, suggesting a different functionality of Brd4 at the genes of ICAM-1 and VCAM-1.

## 4. Discussion

### 4.1. Pretubulysin and Other Depolymerizing MTAs Reduce Inflammatory Processes in ECs by Deregulation of Brd4 in a JNK-Dependent Mechanism

In this project we aimed at assessing the inhibitory potential of the destabilizing microtubule-targeting agent PT and additional MTAs already in clinical use on the interaction of leukocytes with ECs (Figure 11 gives a brief summary for the tested pathways and obtained results). We analyzed the anti-inflammatory effects of PT in vivo by utilizing a murine IMQ-induced psoriasiform dermatitis model and by intravital microscopy of the mouse cremaster muscle. In addition, we tested the adhesion of monocytes to an endothelial monolayer under flow-conditions in vitro. These experiments clearly demonstrated that treatment with PT significantly reduces the inflammatory response and adhesion of leukocytes to the TNF-activated ECs, verifying the anti-inflammatory potential of PT. In addition, we showed that the reduced adhesion of leukocytes is due to an inhibitory effect of PT on the synthesis of the endothelial CAMs ICAM-1 and VCAM-1, which is dependent on the activity of JNK and in part due to JNK-induced deregulation of the histone-acetyltransferase Brd4. Since the other destabilizing MTAs COL and VIN also led to a decreased synthesis of ICAM-1 and VCAM-1, it is to argue that the anti-inflammatory effects of these compounds share the same mechanistical background following the destabilization of the microtubule skeleton. This is also represented by the fact that all tested MTAs decreased the promoter activity of NFκB and AP-1. Studies performed by Wang et al. [36] have already shown that treatment with the Brd-inhibitor JQ1 generally reduces the promoter activity of NFκB. The observed differences in inhibition of the promoter activity by the MTAs might be explained by the different potentials to depolymerize the microtubule skeleton. Over the course of this study, Filipčík et al. [37] have shown that the JNK upstream kinase MAPK kinase kinase 1 (MAP3K1) contains a tubulin-binding domain that interacts with soluble tubulin heterodimers leading to kinase activation. Hence, the strong increase in JNK activity is directly linked to the depolymerization of the tubulin cytoskeleton.

### 4.2. Microtubule Stabilizer Do Not Show an Inhibitory Action on the Leukocyte Adhesion Cascade

Interestingly, the microtubule stabilizer PAC also decreased the promoter activity of NFκB and AP-1, albeit only slightly. This might be explained by the fact that PAC also induced JNK activity, however to a much weaker extent then it could be observed for the depolymerizing MTAs (data not shown). Many phosphatases, e.g., the phosphatase 2α, that counteract the phosphorylation of kinase targets are known to associate with the microtubule cytoskeleton. The increased stabilization of the microtubules after treatment with stabilizing MTAs might thereby decrease the overall phosphatase activity. It is likely, that the overall increased phosphorylation levels outweigh the observed reduction of the NFκB and AP-1 promoter activity due to PAC. The resulting increased CAM levels on the endothelial surface after treatment with PAC confirm this and clearly demonstrate that PAC has no inhibitory potential concerning the leukocyte adhesion cascade.

### 4.3. Depolymerizing MTAs Induce Pro-Inflammatory Signaling Cascades but Disconnect Promoter Action from Constructive Elongation

In general, our analysis of the influence of PT on the TNF-induced NFκB and AP-1 signaling cascades showed a rather pro-inflammatory behavior of this compound, because PT induced the activity of IKK and strongly induced the activity of JNK compared to TNF treatment alone. Previous studies have already demonstrated the pro-inflammatory potential of colchicine in terms of induction of NFκB activity in the absence of pro-inflammatory stimuli [38]. Our experiments showed that extended PT-treatment in the absence of TNF rather induces the expression of NFκB-p65 and AP-1-cJun, which leads to increased nuclear levels of pro-inflammatory transcription factors after induction with TNF (data not shown). The increased pro-inflammatory signaling after PT treatment also leads to significantly increased levels of NFκB-p65 after a short TNF treatment time and a general over-induction of AP-1-cJun enrichment at the promoters of *icam-1* and *vcam-1*. The JNK-induced AP-1-cJun activity appeared to be high at all treatment times, as shown by ChIP-based enrichment and western blot analysis of cJun. In contrast, the enrichment of p65 decreased over time and especially led to reduced enrichment in the promoter of *vcam-1* after extended TNF and PT treatment at times when the overall NFκB-p65 levels were reduced by PT, as shown by NFκB-p65 western blot analysis.

The differences in enrichment might be due to the decreased p65 levels and the presence of only one promoter-proximal NFκB-binding site in the *icam-1* promoter as compared to the *vcam-1* promoter, which contains two adjacent NFκB motifs close to the transcription start site. The fact that the NFκB-p50 protein levels were reduced already 6 h after TNF- and PT-treatment might also suggest a deregulation of the relative p65:p50 levels over time, which might further decrease the activity of NFκB at the promoters of target genes, because the p65:p50 NFκB heterodimer is known to be the most transcriptionally active NFκB constituent. The additional increase of IκB⍺ protein levels after 10 h and 12 h of treatment might also decrease the NFκB activity at these times and might in part be responsible for the decreased nuclear p65 levels at later treatment times. However, studies concerning the Brd-inhibitor JQ1 have already shown a protective function of Brd4 on the activity of NFκB at target genes, with JQ1 inducing the ubiquitination and degradation of NFκB [39]. Hence, it might be possible that the JNK-induced eviction of Brd4 from target genes might also induce ubiquitination and degradation of NFκB after extended treatment with PT. In view of the increased enrichment of NFκB at short treatment times and the overall induction of the JNK-cJun pathway, it is clear that PT rather induces pro-inflammatory signaling cascades, leading to increased enrichment of pro-inflammatory transcription factors at the target genes. However, the observed effects on the CAM mRNA levels after short PT treatment and the JNK-induced inhibition of these genes at later treatment times suggest that transcription factor binding in these genes does not induce transcription. Hence, destabilizing MTAs disconnect the actions at the CAM promoters from constructive elongation by the Pol II.

### 4.4. PT Induces Different Mechanisms of Action Depending on the Treatment Time

The inhibitory influence of PT regarding the constructive CAM transcript elongation is also represented by the decreased Pol II enrichment at the same time points, when the transcription factor enrichment is drastically increased. The enrichment of Brd4 at the genes of ICAM-1 and VCAM-1 after 2 h of treatment appeared to loosely follow the enrichment of NFκB-p65. This is in accordance with the fact that Brd4 binds, next to its general function in binding to acetylated histones, directly to acetylated p65 to enable chromatin opening and activation of CDK9 to release the stalled Pol II and enter constructive elongation. After 6 h of treatment, the PT-induced decrease of the Brd4 enrichment becomes more pronounced compared to the loss of NFκB-p65 at the CAM promoters, marking the beginning of the JNK-dependent deregulation of Brd4 binding to acetylated p65 and association with the pTEFb complex at the CAM genes. Nishiyama et al. [40] have shown that Brd4 is unloaded from chromosomes following depolymerization of microtubules with the microtubule depolymerizer nocodazole and that this unloading mechanism is dependent on the activity of the JNK pathway and residues in the carboxy-terminal domain of Brd4. Due to that it is to assume that the initial increase of Brd4 at the CAM genes simply follows the increased enrichment of NFκB-p65. However, Song et al. [41] have shown that inhibition of JNK prior to 2 h of UV light exposure reduces the UV light damage-induced interaction of Brd4 with pTEFb, indicating the positive regulation of the Brd4:pTEFb interaction due to the activity of JNK. Thus, it might be possible that the activation of JNK initially stimulates the binding of Brd4 to acetylated p65 and association of Brd4 with pTEFb, but reduces Brd4 binding over time. Here, it is worth mentioning that our analysis of the relative mRNA levels of some transcription factors (i.e., cJun, cFos, data not shown) showed a strong induction over the TNF controls at short pre-treatment times with PT, but a significant and time-dependent reduction of the mRNA levels at extended pre-treatment times. In contrast, the mRNA levels of other transcription factors (e.g., IRF-1, data not shown) rather followed an expressional pattern like it could be seen for the CAMs, with initially reduced mRNA levels, which increased at mean treatment times and further declined after extended pre-treatment. While these differences might correlate to a different functionality of Brd4 at different target genes, the elevated levels of some of the transcription factors after short PT-treatment fit well to an initial induction of the activity of Brd4. However, how treatment with MTAs reduce the overall synthesis of ICAM-1 and VCAM-1 after short treatment is difficult to analyze on the basis of our data. Under baseline condition, much of the nuclear pTEFb is kept in an inactive state by association with HEXIM1 and a 7SK small nuclear RNA. Upon cell-stress, like present after depolymerization of the microtubule cytoskeleton, phosphatases like the phosphatase 2α are released from microtubules and enter the nucleus of cells, where they dephosphorylate CDK9 in the pTEFb:HEXIM1:7SK complex leading to the release of the sequestered pTEFb. However, Ouchida et al. [42] have shown that HEXIM1 itself can bind to NFκB-p65 and repress its transcriptional activity. Hence, it is tempting to assume that the initial and JNK-independent decrease in *icam-1* and *vcam-1* expression is due to a cell-stress induced repressional activity of the released HEXIM-1. Further studies should clarify the role of HEXIM1 in the mechanism of repression of the CAMs after short treatment with depolymerizing MTAs.

Interestingly, our analysis of different PT pre-treatment times and 1 h of TNF treatment revealed a super-induction of CAM mRNAs for ICAM-1 and VCAM-1 after a pre-treatment time of 6 h, when the general Pol II enrichment around the *icam-1* and *vcam-1* promoter under constant TNF stimulation was not and the Brd4 enrichment was only slightly negatively influenced by PT-treatment. This super-induction might be explained by the increased enrichment of pro-inflammatory transcription factors in the CAM promoters after only short stimulation with TNF. However, considering the dual mechanism of action induced by PT, the super-induction of mRNA at around 6 h of pre-treatment with PT suggests that the PT-induced inhibition of CAM synthesis at short and extended treatment times are due to alterations in the ability of the cells to enter constructive transcription and that these alterations are largely independent from an extended stimulation with TNF.

### 4.5. The PT-Evoked Down-Regulation of Both CAMs Is Due to Increased JNK Activity but Differs in the Functional Dependency on Brd4

The observed enrichment of Brd4 after inhibition of JNK clearly showed that the negative regulation of Brd4 at target genes after extended treatment with PT is due the elevated activity of JNK. This elevated JNK activity also induces degradation of Brd4, which might correlate with the increased eviction of Brd4 from its target genes and subsequent ubiquitination. However, while the initial levels of Brd4 were similarly enriched for both CAM genes after short treatment, extended treatment lead to a stronger decrease in Brd4 enrichment at the *vcam-1* gene compared to the *icam-1* gene. In addition, knock-down of the total Brd4 levels evoked an increase in the TNF-induced ICAM-1 levels, and PT further increased ICAM-1 levels after Brd4 knock-down. In contrast, knock-down of Brd4 led to the expected decrease of the TNF-induced VCAM-1 levels. This clearly demonstrates a different functionality of Brd4 in the induction of ICAM-1 and VCAM-1. Previous studies have demonstrated that Brd4 can either have an inductive or repressive function depending on the target gene [41,43]. However, since the mRNA and cell surface levels of both CAMs could be recovered after extended PT treatment and JNK inhibition, it is clear that even though ICAM-1 appears to be repressed by Brd4, both genes are negatively regulated by the PT-induced increased JNK activity. This is also demonstrated by the increased repressive H3K9me3 marks in both CAM genes. Song et al. [41] have demonstrated that for genes that are repressed by Brd4, Brd4 knock-down still leads to decreased levels of pTEFb at these target genes, but does not influence the transcriptional activity of Pol II. This indicates the involvement of other factors that confer transcriptional activity in Brd4-repressed gene. However, since the PT-induced effects on both CAM genes are sensitive to inhibition of JNK, it is likely that these factors similarly react to the activity of JNK.

## 5. Conclusions

In conclusion, we demonstrated that PT and other depolymerizing MTAs clearly show an anti-inflammatory potential by inhibiting the synthesis of ICAM-1 and VCAM-1, two crucial components of the leukocyte adhesion cascade, and that this inhibition leads to reduced adhesion of leukocytes to the endothelium. In addition, we showed that the PT-evoked inhibition of the CAM synthesis is the direct consequence of microtubule depolymerization and the resulting activation of JNK, and that this activity negatively regulates the association of Brd4 with target genes. Especially for the Brd4-induced *vcam-1* gene, this leads to a repression of the transcriptional activity. It will be interesting to see how future studies clarify the role of additional factors that confer positive transcriptional elongation in Brd4-repressed genes like ICAM-1 and how these factors relate to the JNK-induced inhibition of these genes after treatment with destabilizing MTAs.

## Figures and Tables

**Figure 1 cells-12-02112-f001:**
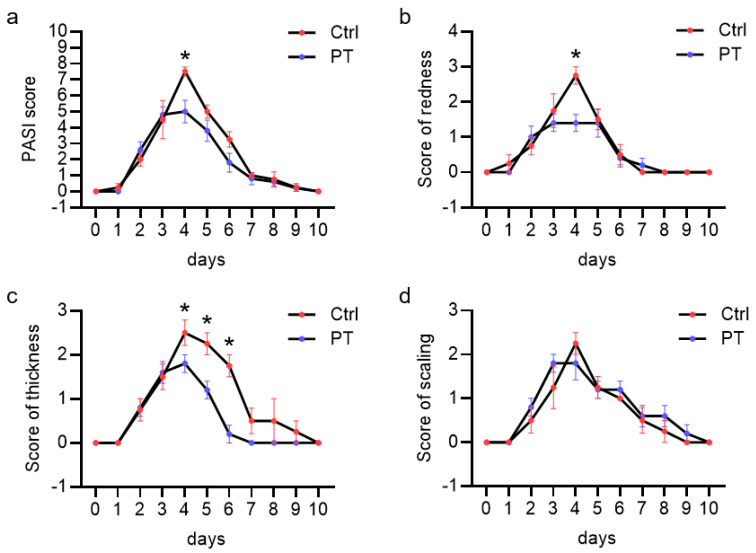
Pretubulysin reduces inflammation in an imiquimod (IMQ)-induced psoriasiform dermatitis mouse model. Mice were treated daily with 62.5 mg IMQ topically applied on the backskin for six consecutive days (starting at day 1). At day 1, 3 and 5, pretubulysin (PT; 1 mg/kg) or vehicle (PBS) was administered (s.c.). (**a**) Cumulative Psoriasis Area Severity Index (PASI) scores as calculated from the individual scores for redness (**b**), thickness (**c**) and scaling (**d**), analyzed daily until 10 days after initial IMQ application. Data are expressed as mean ± SEM. * *p* ≤ 0.05 versus IMQ control for five individual animals.

**Figure 2 cells-12-02112-f002:**
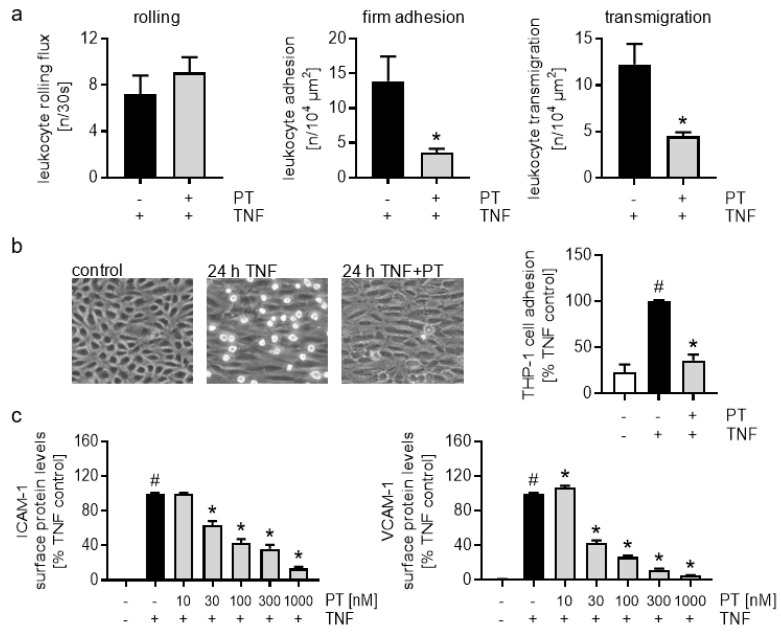
Pretubulysin reduces the interaction of leukocytes with the endothelium in vivo and in vitro and decreases the cell surface levels of CAMs. (**a**) Intravital microscopy of the cremaster muscle. Mice were injected with pretubulysin (PT; 1 mg/kg i.p.) or vehicle. After 30 min, TNF (25 µg/kg) was injected intrascrotally for 4 h, and intravital microscopy was performed to detect intravascular rolling, adhesion and transmigration of leukocytes to the activated endothelium in postcapillary venules of the mouse cremaster muscle. (**b**) THP-1 cell adhesion to endothelial cells under flow condition. HUVECs were cultivated under constant flow (5 dyn/cm2) and pre-treated with PT (300 nM) for 30 min. TNF (10 ng/mL) was added for 24 h, and fluorescence-labeled THP-1 cells were let to adhere at constant shear pressure (0.5 dyn/cm2) for 10 min. Phase-contrast images were taken and the fluorescence signal of the adherent THP-1 cells was measured after cell lysis. (**c**) Flow cytometric analysis of the cell surface protein levels of ICAM-1 (left) and VCAM-1 (right). Confluent HUVECs were pre-treated with different concentrations of PT (10, 30, 100, 300, 1000 nM) for 30 min and activated with TNF (10 ng/mL) for 24 h. (n = 3) Data are expressed as mean ± SEM. # *p* ≤ 0.05 versus control, * *p* ≤ 0.05 versus TNF control.

**Figure 3 cells-12-02112-f003:**
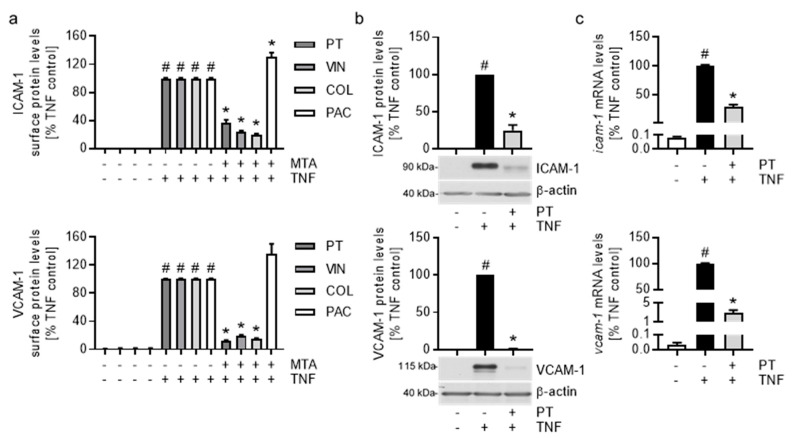
Pretubulysin and other destabilizing MTAs reduce the cell surface protein, total protein and mRNA levels of CAMs involved in the firm adhesion of leukocytes. (**a**) Flow-cytometric analysis of the ICAM-1 (top) and VCAM-1 (bottom) protein cell surface levels. Confluent HUVECs were pre-treated with 300 nM of pretubulysin (PT), vincristine (VIN), colchicine (COL) or paclitaxel (PAC) for 30 min and activated with TNF (10 ng/mL) for 24 h. (**b**) Western blot analysis of the total protein levels of ICAM-1 (top) and VCAM-1 (bottom) in confluent HUVECs pre-treated with pretubulysin (300 nM) for 30 min and activated with TNF (10 ng/mL) for 24 h. The protein levels were normalized on the respective β-actin levels. (**c**) mRNA levels of ICAM-1 (top) and VCAM-1 (bottom). Confluent HUVECs were pre-treated with PT (300 nM) and activated with TNF (10 ng/mL) for 12 h. Results were normalized on the respective *gapdh* levels. (n = 3) Data are expressed as mean ± SEM. # *p* ≤ 0.05 versus control, * *p* ≤ 0.05 versus TNF control.

**Figure 4 cells-12-02112-f004:**
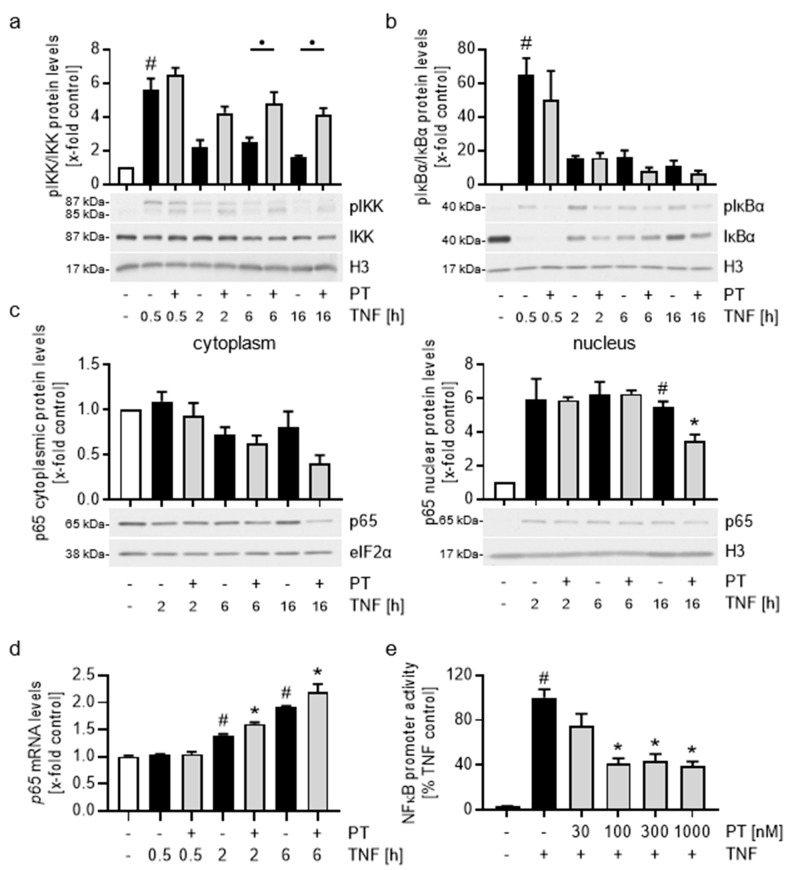
Pretubulysin treatment leads to increased IKK signaling and reduced NFκB promoter activity. Western blot analysis of the IKK (**a**) and IκBα phosphorylation ratio (**b**). Confluent HUVECs were pre-treated with pretubulysin (300 nM) for 30 min and activated with TNF (10 ng/mL) for different durations (0.5, 2, 6, 16 h). Results were normalized on the respective histone 3 (H3) levels. (**c**) Western blot analysis of the NFκB p65 levels in cell fractions. Confluent HUVECs were pre-treated with pretubulysin (300 nM) and activated with TNF (10 ng/mL) for different duration (2, 6, 16 h). Results were normalized on the respective elongation initiation factor 2α (eIF2α) levels (cytoplasmic fraction) or H3 levels (nuclear fraction). (**d**) qPCR analysis of the *p65* mRNA levels. Confluent HUVECs were pre-treated with pretubulysin (300 nM) for 30 min and activated with TNF (10 ng/mL) for different durations (0.5, 2, 6 h). Results were normalized on the respective *gapdh* levels. (**e**) Reporter gene assay of the NFκB promoter activity. HUVECs were transfected with pGL4.32[luc2p/NF-κB-RE/Hygro] and pGL4.74[hRluc/TK] for normalization and pre-treated with different concentrations of pretubulysin (30, 100, 300, 1000 nM) for 30 min and activated with TNF (10 ng/mL) for 6 h. (n = 3) Data are expressed as mean ± SEM. # *p* ≤ 0.05 versus control, * *p* ≤ 0.05 versus TNF control. The bar with dot represents statistical significance over the corresponding data groups with ^•^
*p* ≤ 0.05.

**Figure 5 cells-12-02112-f005:**
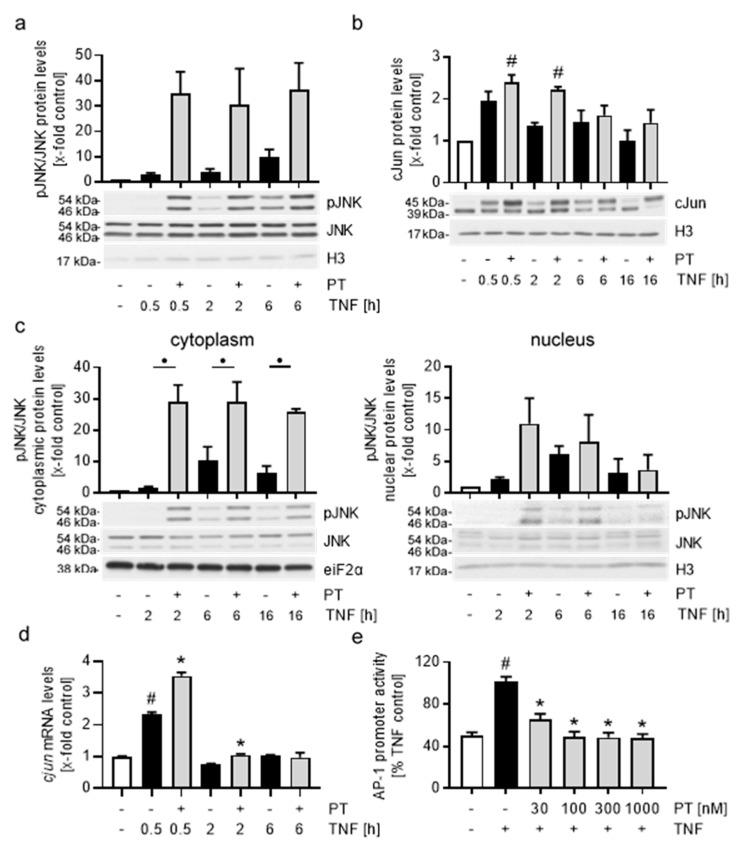
Pretubulysin treatment leads to a strong induction of JNK activity, increased cJun protein levels and a reduced AP-1 promoter activity. Western blot analysis of the JNK phosphorylation ratio (**a**) and total cJun protein levels (**b**). Confluent HUVECs were pre-treated with pretubulysin (PT) for 30 min and activate with TNF (10 ng/mL) for different durations (0.5, 2, 6, 16 h). Results were normalized on the respective histone 3 (H3) levels. (**c**) Western blot analysis of the JNK phosphorylation ratio in cell fractions. Confluent HUVECs were pre-treated with pretubulysin (300 nM) for 30 min and activated with TNF (10 ng/mL) for different duration (2, 6, 16 h). Results were normalized on the respective elongation initiation factor 2α (eIF2α) levels (cytoplasmic fraction) or H3 levels (nuclear fraction). (**d**) qPCR analysis of the *cjun* mRNA levels. Confluent HUVECs were pre-treated with pretubulysin (300 nM) for 30 min and activated with TNF (10 ng/mL) for different durations (0.5, 2, 6 h). Results were normalized on the respective *gapdh* levels. (**e**) Reporter gene assay of the AP-1 promoter activity. HUVECs were transfected with pGL4.44[*luc2p*/AP-1-RE/Hygro] and pGL4.74[*hRluc*/TK] for normalization and pre-treated with different concentrations of pretubulysin (30, 100, 300, 1000 nM) for 30 min and activated with TNF (10 ng/mL) for 6 h. (n = 3) Data are expressed as mean ± SEM. # *p* ≤ 0.05 versus control, * *p* ≤ 0.05 versus TNF control. The bar with dot represents statistical significance over the corresponding data groups with ^•^
*p* ≤ 0.05.

**Figure 6 cells-12-02112-f006:**
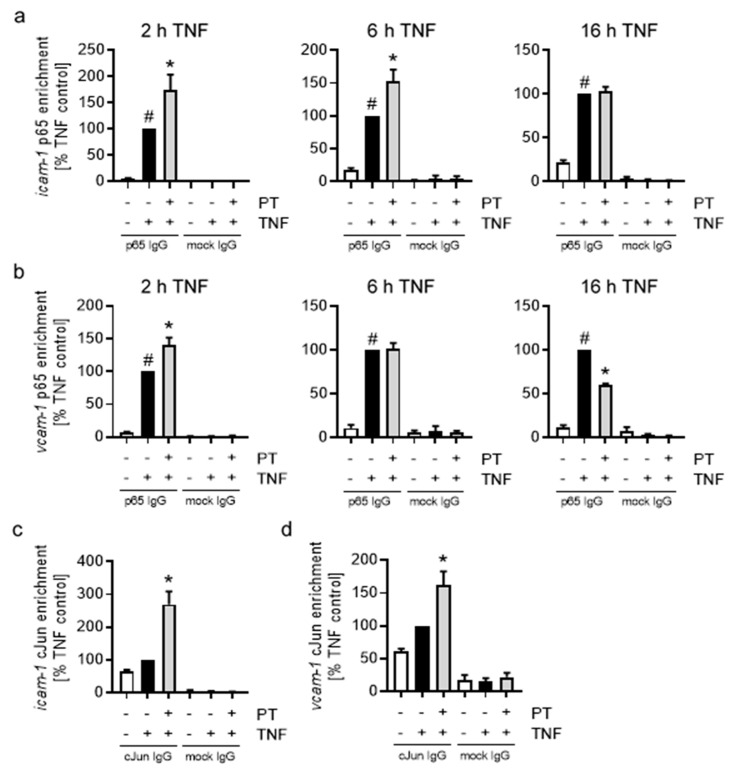
Pretubulysin increases the NFκB-p65 promoter enrichment at short treatment times and leads to a time dependent decrease in p65 enrichment, while strongly inducing the AP-1-cJun enrichment in the *icam-1* and *vcam-1* promoters. Enrichment of NFκB-p65 in the *icam-1* (**a**) and *vcam-1* (**b**) promoters or enrichment of AP-1-cJun in the *icam-1* (**c**) and *vcam-1* (**d**) promoters as shown by chromatin immunoprecipitation (ChIP). Confluent HUVECs were pre-treated with pretubulysin (PT; 300 nM) for 30 min and activated with TNF (10 ng/mL) for a duration of 2, 6, 16 h (**a** and **b**) or 6 h (**c** and **d**). ChIP was performed with a p65 or cJun binding antibody and the respective IgG mock antibody. Results were normalized on the background levels of a sequence of the *rpl13a* 3‘-untranslated region. (n = 3) Data are expressed as mean ± SEM. # *p* ≤ 0.05 versus control, * *p* ≤ 0.05 versus TNF control.

**Figure 7 cells-12-02112-f007:**
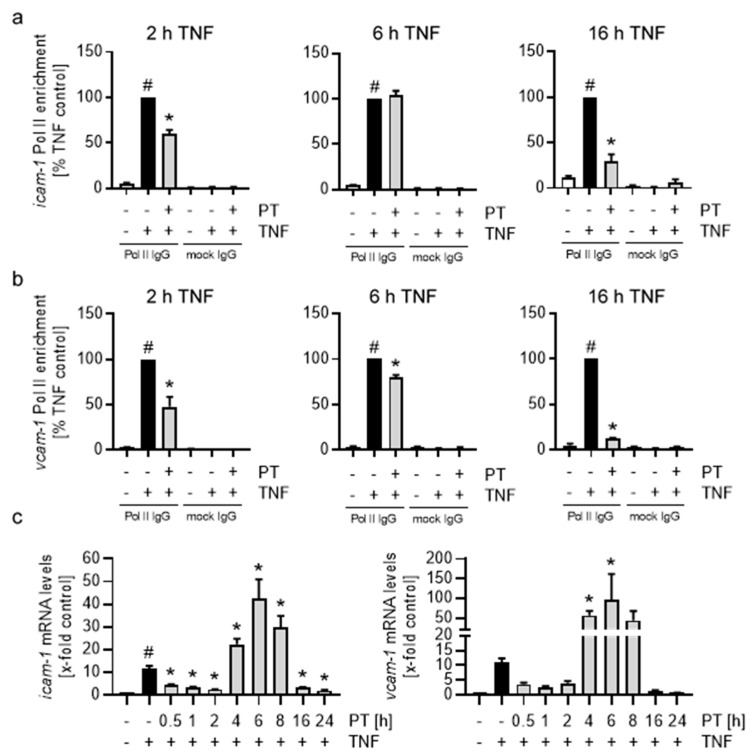
Pretubulysin decreases the RNA polymerase II enrichment at the *icam-1* and *vcam-1* polymerase stalling sites in a time-dependent manner and increases CAM mRNA levels at mean pre-treatment times. Enrichment of the RNA polymerase II in the promoters of *icam-1* (**a**) and *vcam*-1 (**b**) as shown by chromatin immunoprecipitation (ChIP). Confluent HUVECs were pre-treated with PT (300 nM) for 30 min and activated with TNF (10 ng/mL) for different durations (2, 6, 16 h). ChIP was performed with an RNA II polymerase antibody and the respective IgG mock antibody. Results were normalized on the background levels of a sequence of the *rpl13a* 3′-untranslated region. (**c**) qPCR analysis of the mRNA levels of ICAM-1 (left) and VCAM-1 (right). Confluent HUVECs were pre-treated with PT (300 nM) for different durations (0.5, 1, 2, 4, 6, 8, 16, 24 h) and activated with TNF (10 ng/mL) for 1 h. Results were normalized on the respective *gapdh* levels. (n = 3) Data are expressed as mean ± SEM. # *p* ≤ 0.05 versus control, * *p* ≤ 0.05 versus TNF control.

**Figure 8 cells-12-02112-f008:**
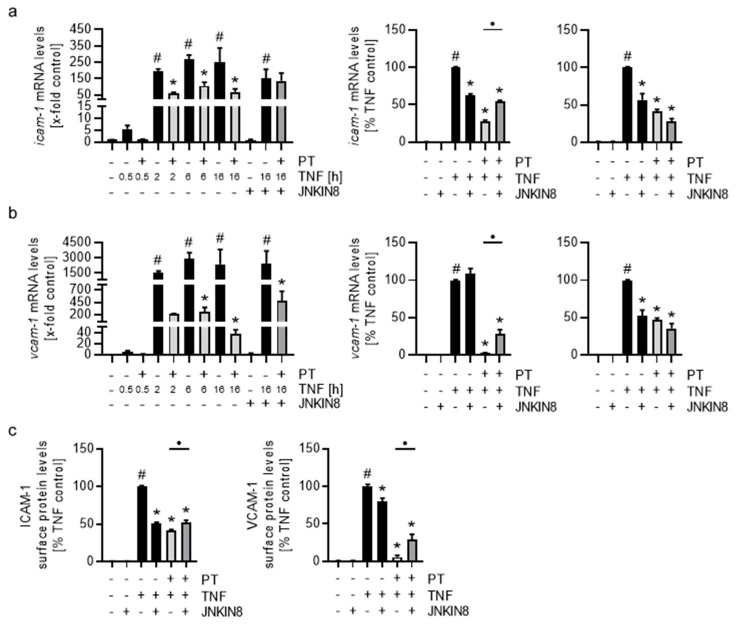
The pretubulysin-induced down-regulation of the *icam-1* and *vcam-1* mRNAs and cell surface protein levels can be recovered by long-term inhibition of JNK activity. qPCR analysis of the *icam-1* (**a**) and *vcam-1* (**b**) mRNA levels in dependency of different TNF induction and long-term JNKIN8 treatment time (left) and percentual representation of long-term TNF induction and JNK inhibition (middle) and short-term TNF induction and JNK inhibition (right). (**c**) Flow cytometric analysis of the effects of JNK inhibition on the pretubulysin-(PT)-induced downregulation of ICAM-1 (left) and VCAM-1 (right) cell surface levels. Confluent HUVECs were pre-treated with pretubulysin (300 nM) for 30 min and activated with TNF (10 ng/mL) for different durations (0.5, 2, 6, 16 h; **a** and **b** left). For inhibition of JNK activity HUVECs were pre-treated with JNKIN8 (5 µM) for 30 min, treated with pretubulysin (300 nM) for further 30 min and activated with TNF (10 ng/mL) for 16 h (**a** and **b** left/middle; long-term JNK inhibition) or 1 h (**a** and **b** right; short-term JNK inhibition) or activated with TNF for 24 h (**c**). For qPCR analysis, results were normalized on the respective *gapdh* levels. (n ≥ 3) Data are expressed as mean ± SEM. # *p* ≤ 0.05 versus control, * *p* ≤ 0.05 versus TNF control. The bar with dot represents statistical significance over the corresponding data groups with ^•^
*p* ≤ 0.05.

**Figure 9 cells-12-02112-f009:**
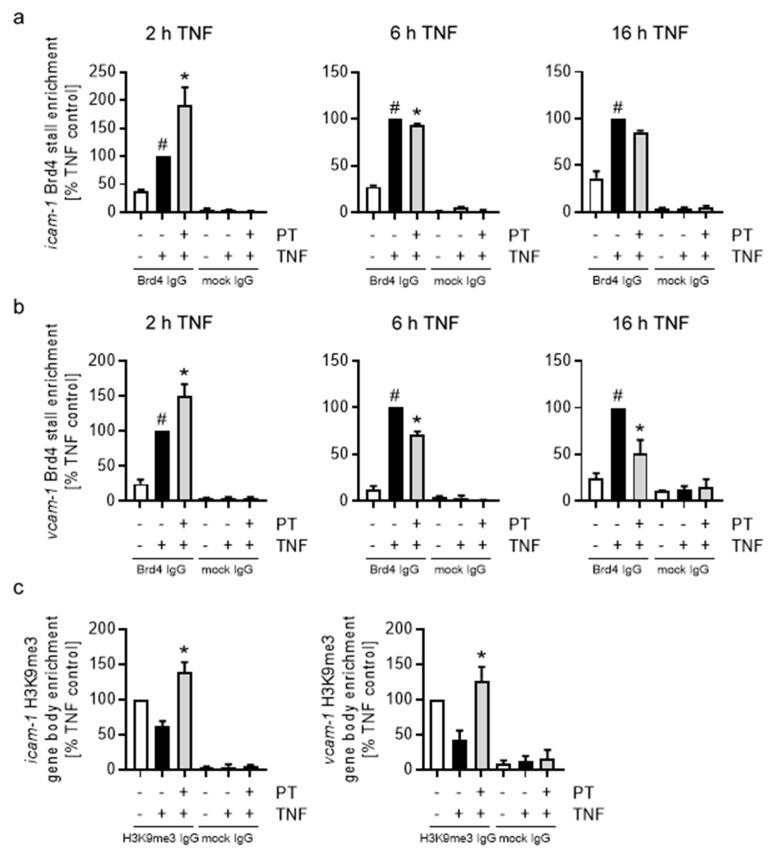
Pretubulysin increases Brd4 enrichment at short treatment times and leads to a time-dependent decrease in Brd4 enrichment at the *icam-1* and *vcam-1* RNA polymerase II stall site, while increasing H3K9me3 enrichment in the gene body after long treatment times. Enrichment of Brd4 at the RNA polymerase stall sites in the *icam-1* (**a**) or *vcam-1* (**b**) gene and H3K9me3 enrichment in the gene body of *icam-1* (**c**, left) and *vcam-1* (**c**, right). Confluent HUVECs were pre-treated with pretubulysin (PT; 300 nM) and activated with TNF (10 ng/mL) for different durations (2, 6, 16 h; **a** and **b**) or for 16 h (**c**) as shown by chromatin immunoprecipitation (ChIP). ChIP was performed with a Brd4 or H3K9me3 antibody and the respective IgG mock antibody. Results were derived based on 2% of input DNA. (n = 3) Data are expressed as mean ± SEM. # *p* ≤ 0.05 versus control, * *p* ≤ 0.05 versus TNF control.

**Figure 10 cells-12-02112-f010:**
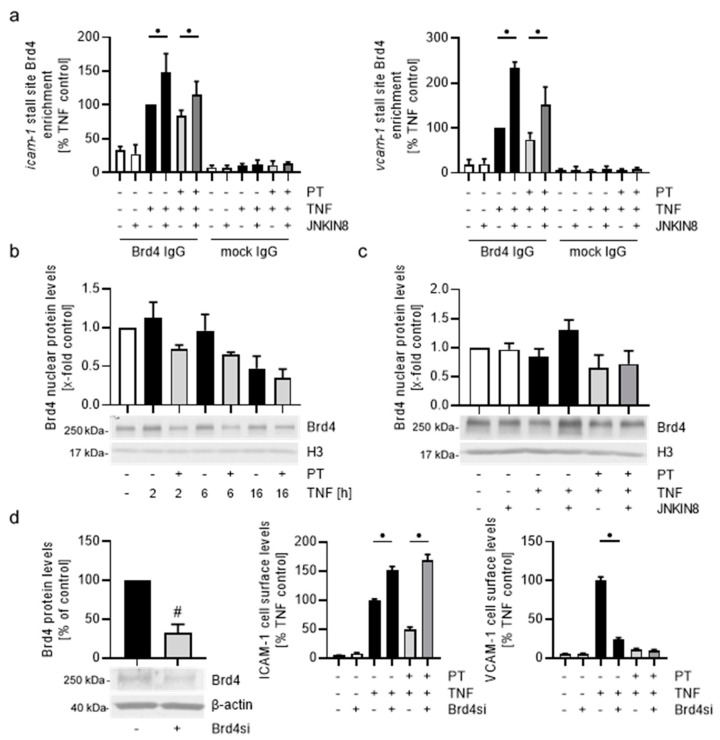
The pretubulysin-induced reduction of Brd4 enrichment in the *icam-1* and *vcam-1* RNA polymerase stalling sites and the PT-induced decrease in Brd4 protein levels can be reversed by inhibition of JNK. (**a**) Chromatin immunoprecipitation (ChIP)-based enrichment of Brd4 at the *icam-1* and *vcam-1* RNA polymerase stalling sites in dependency of JNK inhibition. Confluent HUVECs were pre-treated with JNKIN8 (5 µM) for 30 min, treated with PT (300 nM) for further 30 min and activated with TNF (10 ng/mL) for 6 h. ChIP was performed with an antibody against Brd4 or the respective mock antibody. Results were obtained based on 2% of input. (**b**) Western blot analysis of the Brd4 protein levels in the nuclear fraction and (**c**) nuclear Brd4 levels in dependency of JNK inhibition. Confluent HUVECs were pre-treated with pretubulysin (PT; 300 nM) for 30 min and activated with TNF (10 ng/mL) for different durations (2, 6, 16 h; **b**) or pre-treated with JNKIN8 (5 µM) for 30 min, treated with PT (300 nM) for further 30 min and activated with TNF (10 ng/mL) for 6 h (**c**). Results were normalized on the respective histone 3 (H3) levels. (**d**) Western blot analysis of the total Brd4 protein levels (left) or flow-cytometric analysis of the ICAM-1 and VCAM-1 cell surface levels (middle and right) after Brd4 knock-down. Sub-confluent HUVECs were transfected with 100 nM of Brd4 siRNA or non-targeting siRNA and incubated for 48 h, pre-treated with PT (300 nM) for 30 min and activates with TNF (10 ng/mL) for 24 h and were used for flow-cytometry or western blot analysis. Western blot results were normalized on the respective β-actin levels (**d**, left). (n = 3) Data are expressed as mean ± SEM. The bar with dot represents statistical significance over the corresponding data groups with ^•^
*p* ≤ 0.05.

**Figure 11 cells-12-02112-f011:**
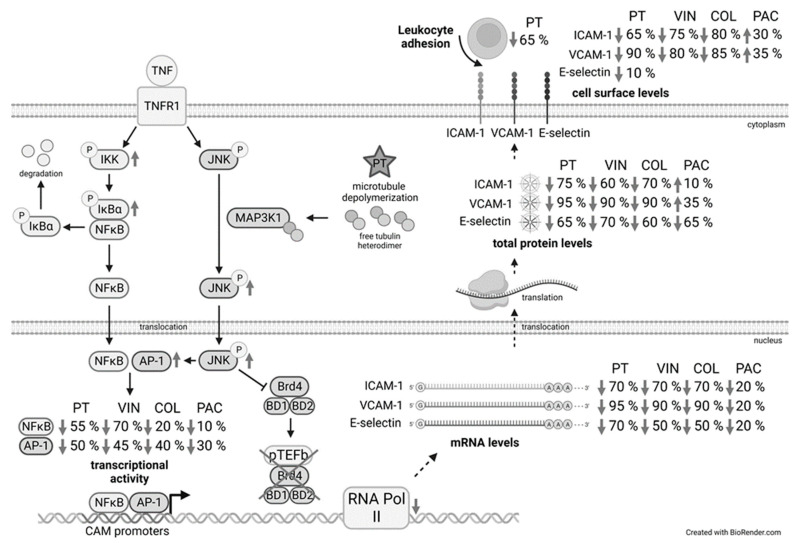
Schematic representation of the effects of PT and VIN, COL and PAC in endothelial cells (after treatment with 300 nM; percent decrease or increase in relation to the respective TNF control). Binding of TNF to the TNF receptor leads to intracellular activation of IKK and JNK. Treatment with PT increases the phosphorylation of IKK and subsequent degradation of IκBα, leading to an unimpaired NFκB translocation. Microtubule depolymerization due to PT increases the concentration of free tubulin heterodimers. MAP3K1, which has been shown to be activated by free-tubulin dimers, significantly increases JNK activity. The strongly activated JNK activates cJun in the AP-1 transcription factor, leading to the autoinduction of cJun transcription by activated cJun, increasing the cJun protein levels. NFκB-p65 and AP-1-cJun both bind to the promoters of the CAMs, but the transcriptional activity of both transcription factors is significantly reduced after treatment with the depolymerizing MTAs. Under TNF stimulation, Brd4 (which contains two bromodomains BD1 and BD2) associates with pTEFb in order to release the stalled RNA polymerase II in the non-translated regions downstream of the transcription start sites of the CAM promoters. The increased JNK activity negatively regulates the association of Brd4 with chromatin and likely prevents proper formation of the pTEFb-Brd4 complex. In addition, treatment with PT also reduces the levels of the RNA polymerase in the promoters and stalling regions of the CAMs. The effects of the depolymerizing agents result in decreased expression of CAM mRNA and total CAM protein levels as well as cell surface levels. The decreased CAM levels also significantly reduce the adhesion of leukocytes to the endothelial cells, thereby reducing leukocyte infiltration.

## Data Availability

Data is contained within the article and Appendix A.

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
