# Peer review of "The Microtubule-Targeting Agent Pretubulysin Impairs the Inflammatory Response in Endothelial Cells by a JNK-Dependent Deregulation of the Histone Acetyltransferase Brd4"

_cells, 2023, doi:10.3390/cells12162112_

Round 1

Reviewer 1 Report

Review

The microtubule-targeting agent pretubulysin impairs the in- 2 flammatory response in endothelial cells by a JNK-dependent 3 deregulation of the histone acetyltransferase Brd4

by Primke et al. submitted to Cells in July 2023

Cells-2531440

 The authors show that pretubulysin, a microtubule targeting agent, regulates ICAM-1 and VCAM-1 expression through JNK-induced deregulation and degradation of the histone acetyltransferase brodomain-containing protein4 in ECs.

 The topic of the investigation is interesting and biologically relevant. The study is well structured, the methods are extensive and well described and the results are convincing. In my opinion, the manuscript tends to be too long and contains too many figures. As a result, the main findings are somewhat lost.

Minor Objections:

-        Negative controls for all the RT-PCRs are missing (non-RT controls, NTCs, and genomic DNA). Only these controls (especially the non-RT controls) ensure the specificity of the results.

-        How were the inhibitors dissolved? What exactly do the controls in the inhibitor experiments represent?

-        What is known about the biological half-life of pretubulysin, particularly in mice?

-        It was mentioned that higher concentrations of pretubysin have a negative impact on cell viability. How was this determined?

-        Fractionation experiments lack protein detection of H3 for the cytosolic lysate and a protein present exclusively in the cytoplasm for the nuclear lysate to indicate purity of fractionation.

-        Why do the authors use H3 as the housekeeping gene for most western blots? Are they nuclear lysates or whole cell lysates?

-        It looks like there is no basal expression of ICAM-1 in almost all experiments, is this correct? Or should the axis be changed to see any expression - also compared to the TNF control?

-        Why is there no error bar in some controls, such as Fig. 4C?

-        c-Jun and JNK have two bands. Which of them was analyzed denstometrically?

-        All western blot images lack kDa information.

-        The formatting still needs to be corrected for some text sections, e.g. in Figure 6.

-        Perhaps the authors can say more precisely at which position within the examined genes the primers used in ChIP-RT-qPCR bind.

-        The transfection efficiency of siRNA was checked by western blot as shown in Fig. 10d. Why are the bands so weak, even in the control. It is difficult to detect differences here.

-        I recommend adding a schematic figure showing the main results and mechanisms involved.

Author Response

The detailed rebuttal letter is for both reviewers and can be found as a separate PDF document attachment.

Reviewer 2 Report

The authors studied the role of pretubulysin in inhibiting inflammation and the pathways involved. This article contains a lot of data. The introduction provides an overview of the pathways involved in inflammation. In the material and methods section, line 94, the PMID of the article is given to cite the synthesis of the product tested, whereas everywhere else the number corresponding to the reference is given. On line 126, the abbreviation IMQ is given, but no explanation is given in the text. The IMQ reference arrives on line 407. Line 217, a space is missing between 1000 and "g". For part 2.10 qPCR: it might be better to make a table to indicate the primers, as this would be easier for a reader who would like to use them. The same applies to part 2.13.

The results are well presented. What would be interesting for the discussion and conclusion, and also helpful for the introduction, would be to make a summary diagram of the pathways involved in inflammation and to indicate the effect of the different products tested.

Author Response

(The authors gave the same response as above.)
